# MethCORR modelling of methylomes from formalin-fixed paraffin-embedded tissue enables characterization and prognostication of colorectal cancer

Trine B. Mattesen [1], Mads H. Rasmussen [1], Juan Sandoval[2,3], Halit Ongen [4], Sigrid S. Árnadóttir[1], Josephine Gladov [1], Anna Martinez-Cardus [5,6], Manuel Castro de Moura[7], Anders H. Madsen[8], Søren Laurberg[9], Emmanouil T. Dermitzakis [4], Manel Esteller [10,11,12,13], Claus L. Andersen [1,14✉] & Jesper B. Bramsen [1,14✉]

Transcriptional characterization and classification has potential to resolve the inter-tumor heterogeneity of colorectal cancer and improve patient management. Yet, robust transcriptional profiling is difficult using formalin-fixed, paraffin-embedded (FFPE) samples, which complicates testing in clinical and archival material. We present MethCORR, an approach that allows uniform molecular characterization and classification of fresh-frozen and FFPE samples. MethCORR identifies genome-wide correlations between RNA expression and DNA methylation in fresh-frozen samples. This information is used to infer gene expression information in FFPE samples from their methylation profiles. MethCORR is here applied to methylation profiles from 877 fresh-frozen/FFPE samples and comparative analysis identifies the same two subtypes in four independent cohorts. Furthermore, subtype-specific prognostic biomarkers that better predicts relapse-free survival (HR = 2.66, 95%CI [1.67–4.22], $P$ value < 0.001 (log-rank test)) than UICC tumor, node, metastasis (TNM) staging and microsatellite instability status are identified and validated using DNA methylation-specific PCR. The MethCORR approach is general, and may be similarly successful for other cancer types.

[1] Department of Molecular Medicine, Aarhus University Hospital, 8200 Aarhus, Denmark. [2] Epigenomic Unit, Health Research Institute La Fe (ISSLaFe), Valencia, Spain. [3] Biomarker and precision medicine Unit, Health Research Institute La Fe (ISSLaFe), Valencia, Spain. [4] Genetic Medicine and Development, University of Geneva Medical School-CMU, 1 Rue Michel-Servet, 1211 Geneva, Switzerland. [5] Badalona Applied Research Group in Oncology (B-ARGO), Germans Trias i Pujol Research Institute (IGTP), Badalona, Barcelona, Catalonia, Spain. [6] Medical Oncology Service, Institute Catalan of Oncology (ICO), Badalona, Barcelona, Catalonia, Spain. [7] Josep Carreras Leukaemia Research Institute (IJC), Badalona, Barcelona, Catalonia, Spain. [8] Department of Surgery, Hospitalsenheden Vest, 7400 Herning, Denmark. [9] Colorectal Surgical Unit, Department of Surgery, Aarhus University Hospital, 8200 Aarhus, Denmark. [10] Josep Carreras Leukaemia Research Institute (IJC), Badalona, Barcelona, Catalonia, Spain. [11] Centro de Investigacion Biomedica en Red Cancer (CIBERONC), Madrid, Spain. [12] Institucio Catalana de Recerca i Estudis Avançats (ICREA), Barcelona, Catalonia, Spain. [13] Physiological Sciences Department, School of Medicine and Health Sciences, University of Barcelona (UB), Barcelona, Catalonia, Spain. [14]These authors contributed equally: Claus L. Andersen, Jesper B. Bramsen ✉email: Cla@clin.au.dk; Bramsen@clin.au.dk

Colorectal cancer (CRC) is a disease with extensive inter-patient heterogeneity, both molecularly and histopathologically, which cannot be resolved by current clinical methods. Despite a continuous refinement of the UICC tumor, node, metastasis (TNM) staging system to measure disease extent and define prognosis, the disease outcome still varies considerably even for patients with the same tumor stage. Therefore, new factors that can more precisely stratify patients into different risk categories are clearly warranted[1].

Recent attempts to resolve CRC heterogeneity and improve prognostication include molecular subclassification and characterization based on transcriptional profiling[2–4]. Consensus molecular subtype (CMS) classification stratifies CRC into four subtypes CMS 1–4 with distinct biology and histopathological features[2]. Still, the CMS taxonomy itself has limited prognostic power for therapeutic decision-making[5]. To address this, we previously combined transcriptional subtyping with subtype-specific prognostic biomarkers to improve prognostication beyond TNM staging in retrospective cohorts[3]. This indicated a clinical potential of using molecular classification and subtype-specific biomarkers as a complement to TNM staging for prognostication. Furthermore, it highlighted the importance of archived tumor material for biomarker discovery and pre-clinical validation.

The strategies for transcriptional classification and subtype-specific prognostication were developed by, and still primarily rely on, profiling high-quality RNA purified from fresh-frozen (FF) tissue. However, high-quality RNA is often not recovered from the formalin-fixed, paraffin-embedded (FFPE) tissue that is routinely archived in the clinic. This can preclude confident transcriptional profiling and hereby complicate clinical testing of molecular classification and exploratory analysis in well-annotated, archival FFPE material[5–9]. The clinical popularity of FFPE tissue is unlikely to change as it forms the basis for histopathological diagnoses and is a convenient, cost-effective preservation method. For wide utility, strategies for molecular classification, characterization, and prognostication should therefore be compatible with FFPE tissue.

Strategies based on DNA rather than RNA profiling may be a way forward. DNA is considered less sensitive to degradation than RNA in FFPE samples[10,11] and enzymatic strategies for DNA repair have greatly improved the analysis of FFPE DNA[12–15]. A strategy for robust analysis of clinical and archival FFPE samples could involve DNA methylation as highly concordant DNA methylation profiles are produced from matched FF and FFPE tissues when using the Illumina Infinium Human-Methylation Beadchip technology[14,16,17]. In addition, many biological traits, such as RNA expression and cell-type identity, are associated with specific and robust DNA methylation patterns in the genome[18,19]. This suggests that both gene expression and cell-type information may be extracted from DNA methylation profiles of FFPE samples and used for molecular classification and prognostication, as an alternative to RNA profiling. Furthermore, conversion of methylation profiles into a gene-centric expression format would allow molecular analysis of FF and FFPE samples using the plethora of bioinformatics tools, databases, and signatures established for RNA expression analysis.

Motivated by this, we here develop MethCORR, an approach, which identifies genome-wide correlations between gene expression and DNA methylation and use this to obtain gene expression and cell-type information in independent samples from their DNA methylation profiles. In homogenous cell preparations, associations between gene expression and DNA methylation have been observed only for a small fraction of genes when analyzing local promoters, gene bodies, or nearby enhancers[20–22]. We hypothesize that genome-wide correlation analysis will identify far more associations and that these will include both functional gene-regulatory interactions and indirect associations e.g. between cell-type-specific RNA expression and cell-type-specific DNA methylation. We here show that MethCORR, independent of whether the methylomes were produced from FFPE or FF tissues, allows expression information to be inferred for a large number of genes (>11,000). Consequently, MethCORR enables a plethora of molecular analyses to be performed on otherwise difficult-to-analyze FFPE tissues e.g. tumor characterization, tumor classification, and interpretation of expression signatures to derive DNA methylation-based biomarkers. Hereby MethCORR also provides a path for improved, subtype-specific prognostication of CRC using clinical FFPE samples.

## Results

**MethCORR infers RNA expression from DNA methylation.** Here we developed the MethCORR approach that, by mapping genome-wide correlations between RNA expression and DNA methylation in FF samples, can infer gene expression information in unrelated samples from their DNA methylation profiles. Correlations were identified genome-wide using matching RNA expression and 450K methylation data (methylation $\beta$-values) from 394 FF CRC samples of The Cancer Genome Atlas (TCGA) Project, denoted the COREAD cohort (Fig. 1a and Supplementary Fig. 1a; Supplementary Table 1 and Supplementary Data 1). The cohort was divided into two discovery sets (each $n = 158$) in which genome-wide correlation analysis was performed independently and one validation set ($n = 78$; Fig. 1a). Our analysis identified positively and negatively expression-correlated CpGs (Spearman's correlation $P$ value < 0.01) overlapping in the two discovery sets for 17,776 of 20,530 genes (Fig. 1a). The majority of the genes without expression-correlated CpGs were non-expressed (Supplementary Fig. 1b). To derive gene expression information for these 17,776 genes, we selected up to 200 CpGs whose methylation level were most negatively ($\leq$100 sites) and positively ($\leq$100 sites) correlated with its expression (Fig. 1a). The methylation levels of these expression-correlated CpGs were used to calculate a MethCORR score (MCS) for each gene (formula in Fig. 1b) and simple linear and polynomial regression modeling was used to identify genes with good correlations between MCSs and measured RNA expression (Fig. 1a). Models were established in the discovery sets by ten times tenfold cross validation and selected using root mean square error (RMSE) as a measure of model fit. We found good inter-sample correlations for 16,248 genes in the discovery sets ($R^2 > 0.16$) and confirmed these for 11,222 genes in the validation set (gene model performances in Supplementary Data 2; Supplementary Fig. 1c–e). The 11,222 genes were denoted MethCORR genes and the expression-correlated CpGs of these define the COREAD MethCORR matrix ($\leq$200 CpGs × 11,222 genes; Supplementary Data 3) that was used for calculation of MCSs from DNA methylation profiles of all samples analyzed in this study (Fig. 1c). We also investigated if RNA expression was better modeled using the $\leq$200 expression-associated CpGs for each gene directly, instead of using MCSs, but found no improvement in overall performance ($R^2$ and RMSE; Supplementary Fig. 1f). Similarly, adding age and gender information to MCS-based models did not improve overall performances (Supplementary Fig. 1g). This likely reflect that CRC-induced methylation changes are much greater than the subtler effects of age and gender in normal tissues[23]. Still, MethCORR captures gender-specific expression by including CpGs located on chromosome X and Y in the MethCORR matrix. Accordingly, known gender-specific RNAs exhibited gender-specific inferred RNA expression (Supplementary Fig. 1h).

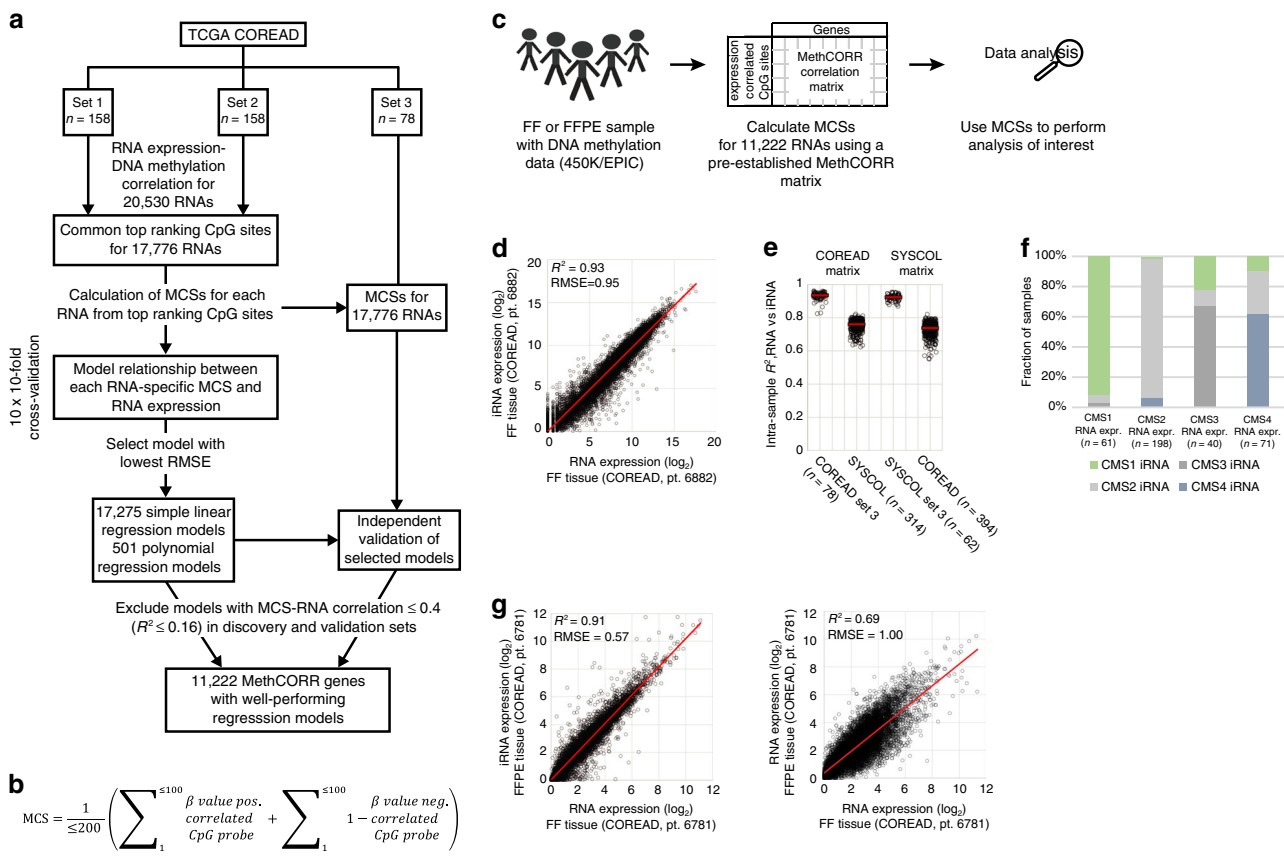

**Fig. 1 Development of the MethCORR approach, MethCORR scores, and inferring of RNA expression. a** Overview of MethCORR development in the COREAD cohort using matched RNA-sequencing and 450K methylation data. The cohort was divided into two discovery sets (each $n = 158$) and one validation set ($n = 78$). Genome-wide RNA expression-DNA methylation correlations were identified in each discovery set and shared top expression-correlated CpGs for each RNA were selected ($\leq 100$ positively and $\leq 100$ negatively correlated CpGs; Spearman's correlation $P$ value < 0.01). A MCS was calculated for each gene using DNA methylation $\beta$-values of expression-correlated CpGs and the formula given in (**b**). RNA expression of each gene was modeled from its MCS using simple linear- and polynomial regression models and $10 \times 10$-fold cross validation in set $1 + 2$. Simple linear models were selected for all RNAs except when polynomial models exhibited a $\geq 5\%$ decrease in RMSE values (Supplementary Fig. 1d). Only models with $R^2 > 0.16$ between inferred RNA (iRNA) and observed RNA expression in both the discovery set and the independent validation set 3 was kept for further analysis ($n = 11,222$, termed MethCORR genes). **b** Formula for calculating MCSs from DNA methylation $\beta$-values. **c** Overview of MethCORR applications. Fresh-frozen (FF)/FFPE CRC samples with 450K/EPIC methylation profiles can be applied to the MethCORR matrix for calculation of MCSs and iRNA expression. **d** Scatterplot showing intra-sample correlation between iRNA and RNA expression in a representative COREAD validation sample. **e** Plot showing $R^2$ of intra-sample iRNA and RNA expression correlations for all samples of the COREAD validation set 3 and SYSCOL cohort when using the COREAD-derived MethCORR matrix (left) and for all samples of the SYSCOL validation set 3 and COREAD cohort when using the SYSCOL-derived MethCORR matrix (right). **f** Histogram showing overlap in CMS subtype predictions in COREAD CRC samples using RNA expression or iRNA expression for classification. **g** Scatterplot showing intra-sample correlation between iRNA (left) or RNA expression (right) from a FFPE sample and RNA expression in a matched fresh-frozen COREAD sample.

Next, we investigated characteristics of the MethCORR genes included in the MethCORR matrix. MethCORR genes exhibited greater variation in RNA expression (Supplementary Fig. 2a), were more frequently dysregulated in cancer vs. normal mucosa (Supplementary Fig. 2b) and encompassed relatively fewer household genes (Supplementary Fig. 2c) than the set of genes not included in the MethCORR matrix. Importantly, the MethCORR genes exhibited the same stroma score distribution as the full set of genes (Supplementary Fig. 2d). This indicates that MethCORR maintains the ability to characterize both traits of the cancer cells and the surrounding stroma. The established MCS regression models were next used to calculate inferred RNA (iRNA) expression for MethCORR genes in the validation samples of the COREAD cohort (set 3) and in an independent Danish CRC cohort, denoted SYSCOL[3]. We found a high intra-sample correlation between measured RNA and iRNA expression in the COREAD validation samples (median $R^2 = 0.93$ (range = 0.82–0.96); Supplementary

Data 4) and SYSCOL samples (median $R^2 = 0.76$ (range = 0.62–0.82); Fig. 1d–e; Supplementary Data 5). To evaluate the robustness of MethCORR to differences between cohorts, we repeated the entire MethCORR discovery and validation process using the SYSCOL cohort to construct a SYSCOL MethCORR matrix, derive MCSs, and to infer iRNA expression (Fig. 1a; Supplementary Data 6–7). Again, we found high intra-sample correlations between observed RNA and iRNA expression (SYSCOL set 3, median $R^2 = 0.92$ (range = 0.87–0.95); COREAD median $R^2 = 0.74$ (range = 0.55–0.82); Fig. 1e; Supplementary Data 4 and 5). We speculated that the moderate decrease in $R^2$ between cohorts was caused by differences in RNA quantification methods rather than the MethCORR approach. In support, comparative analysis of COREAD validation samples using normalized RNA expression data from the UCSC XENA database[24] and the National Cancer Institute (NCI) genomic database commons (GDC)[25] confirmed that MethCORR iRNA-RNA correlations were not

**Table 1** $R^2$ and RMSE for intra-sample correlations between MethCORR inferred RNA expression (iRNA), RNA expression, or MCS in FFPE samples and RNA expression or MCS in matched fresh-frozen tissue.

| TCGA COREAD patient Id | $R^2$ iRNA (FFPE) vs. RNA (FF) | $R^2$ RNA (FFPE) vs. RNA (FF) | $R^2$ MCS (FFPE) vs. MCS (FF) | RMSE iRNA (FFPE) vs. RNA (FF) | RMSE RNA (FFPE) vs. RNA (FF) | RMSE MCS (FFPE) vs. MCS (FF) |
|---|---|---|---|---|---|---|
| Pt. 6650 | 0.94 | 0.87 | 1.00 | 0.47 | 0.69 | 0.04 |
| Pt. 5659 | 0.92 | 0.74 | 1.00 | 0.54 | 1.08 | 0.03 |
| Pt. 5661 | 0.92 | 0.67 | 0.99 | 0.54 | 1.25 | 0.03 |
| Pt. 5665 | 0.91 | 0.72 | 0.98 | 0.57 | 1.02 | 0.04 |
| Pt. 6781 | 0.91 | 0.69 | 0.98 | 0.54 | 1.00 | 0.03 |
| Pt. 6780 | 0.90 | 0.81 | 0.99 | 0.60 | 0.82 | 0.03 |
| Pt. 2684 | 0.88 | 0.67 | 0.98 | 0.65 | 1.03 | 0.04 |
| Pt. 3810 | 0.87 | 0.70 | 1.00 | 0.66 | 0.98 | 0.02 |
| Pt. 5656 | 0.80 | 0.63 | 0.98 | 0.83 | 1.11 | 0.07 |

lower than if applying two different RNA normalization strategies to the same samples (Supplementary Fig. 2e).

In accordance with the high intra-sample correlations between measured RNA and iRNA expression, we found a good overlap in CMS (84% agreement) and CRC intrinsic subtype (CRIS; 75% agreement) predictions when using the measured RNA or iRNA expression as input (Fig. 1f and Supplementary Fig. 2f).

In situations where high-quality RNA is not obtainable, iRNA expression may provide better estimates of gene expression than RNA sequencing, as even moderate declines in RNA quality can lead to unreliable expression profiles[26,27]. Indeed, samples with the lowest correlation between measured RNA and iRNA expression had significantly lower RNA quality than high correlation samples ($P$ value < 0.0001, Wilcoxon rank sum (WRS) test; Supplementary Fig. 2g). In contrast, no equivalent drop in 450K methylation data quality was observed (Supplementary Fig. 2g). Compromised RNA quality is inherent to FFPE tissue[10,11]. In agreement, analysis of nine COREAD samples with available RNA sequencing and 450K methylation profiles from matched FF and FFPE tissues identified higher intra-sample $R^2$'s between FF RNA sequencing and FFPE iRNA profiles (median $R^2 = 0.91$ (range: 0.80–0.94)) than between FF and FFPE RNA-sequencing profiles (median $R^2 = 0.7$ (range: 0.63–0.87); $P$ value < 0.001, WRS test; Fig. 1g and Table 1; Supplementary Data 8–11 and Supplementary Table 2). MCSs from matched FFPE and FF samples were even higher correlated (median $R^2 = 0.98$ (range: 0.98–1.00); Table 1), which likely reflect that 450K methylation profiles were themselves highly correlated (median $R^2 = 0.96$ (range: 0.94–0.98); Supplementary Fig. 2h), as reported previously[14,16,17]. Additional evidence came from principal component analysis (PCA). Here samples clustered according to preservation method when analyzing FF and FFPE RNA-sequencing profiles together, whereas samples clustered more according to patient ID when analyzing RNA profiles of FF samples together with iRNA or MCS profiles of FFPE samples (Supplementary Fig. 2i).

Collectively, this showed that MethCORR expression measures (MCSs and iRNAs) can be inferred from DNA methylation for a large number of genes, even when methylation data are based on FFPE tissue.

**MethCORR identifies two subtypes in FF and FFPE cohorts.** We next investigated if inferred expression profiles allow uniform subtype discovery and characterization of both FF and FFPE cohorts using bioinformatics strategies normally reserved for FF samples with high-quality RNA expression profiles. As input, we employed MCS profiles as they strengthen the focus on cancer cell-related traits during subtype discovery as compared with RNA and iRNA profiles (Supplementary Fig. 3a, b). Subtype discovery by non-negative matrix factorization (NMF)-based

consensus clustering was performed in TNM stage II–III COREAD and SYSCOL samples with available 450K methylation data and in two independent FFPE TNM stage II–III cohorts, denoted FFPE1 and FFPE2 (Supplementary Table 1 and Supplementary Data 12). Our focus was on stage II–III patients, which are most relevant for prognostic biomarker identification due to their heterogeneous prognosis[1]. Two MethCORR subtypes, CRC1 and CRC2, were identified in all four cohorts (Supplementary Fig. 3c) and Submap analysis[28] confirmed the correspondence between the CRC1 and CRC2 subtypes in the different cohorts (Supplementary Fig. 3d; FDR < 0.05). In agreement, samples clustered according to subtype in a PCA of all four CRC cohorts together, irrespectively of their preservation-type status (Supplementary Fig. 3e). We next performed comparative subtype characterization in all cohorts, which indicated that CRC1 and CRC2 differed in terms of DNA methylation, chromosomal instability, and stromal/immune cell activity (Fig. 2a and Supplementary Fig. 3f). These are well-known characteristics for the serrated/microsatellite instability status (MSI) and conventional CRC pathways, respectively, pointing to a biological relevance of the MethCORR subtypes.

Further subtype characterization was performed using pre-ranked gene set enrichment analysis (GSEA)[29]. Initially, we investigated if similar gene set enrichments were identified when using MCSs vs. RNA expression as input (Fig. 2b) or when MCSs were derived from FF vs. FFPE samples (Fig. 2c). Indeed, a high concordance was observed between normalized enrichment scores for most gene sets in both situations, supporting that expression-correlated MCSs can substitute RNA expression and enable analysis of FFPE tissue. MCS-based GSEA of each cohort uniformly showed that the CRC1 subtype was enriched in gene sets associated with immune- and stromal processes/cell types such as inflammation, epithelial-mesenchymal transition (EMT), cancer-associated fibroblasts (CAFs), and T/B cells (Fig. 2d and Supplementary Table 3). Furthermore, CRC1 was enriched in gene sets associated with positive MSI-, CIMP-, and serrated CRC-status, whereas CRC2 tumors were enriched in gene sets associated with conventional CRC and a more undifferentiated cell status (Fig. 2d and Supplementary Table 3). Similar results were obtained for the two FF cohorts when using RNA expression as input, rather than MCSs (Fig. 2d). Despite biological differences, no difference in relapse-free survival (RFS) was observed between CRC1 and CRC2 (Fig. 2e).

Collectively, these results demonstrate that MethCORR allows uniform discovery and characterization of biologically relevant CRC subtypes in FF and FFPE samples using well-established bioinformatics tools.

**A MethCORR map characterizes CRC subtypes.** By analysis of expression-correlated CpGs in the MethCORR matrix, we found

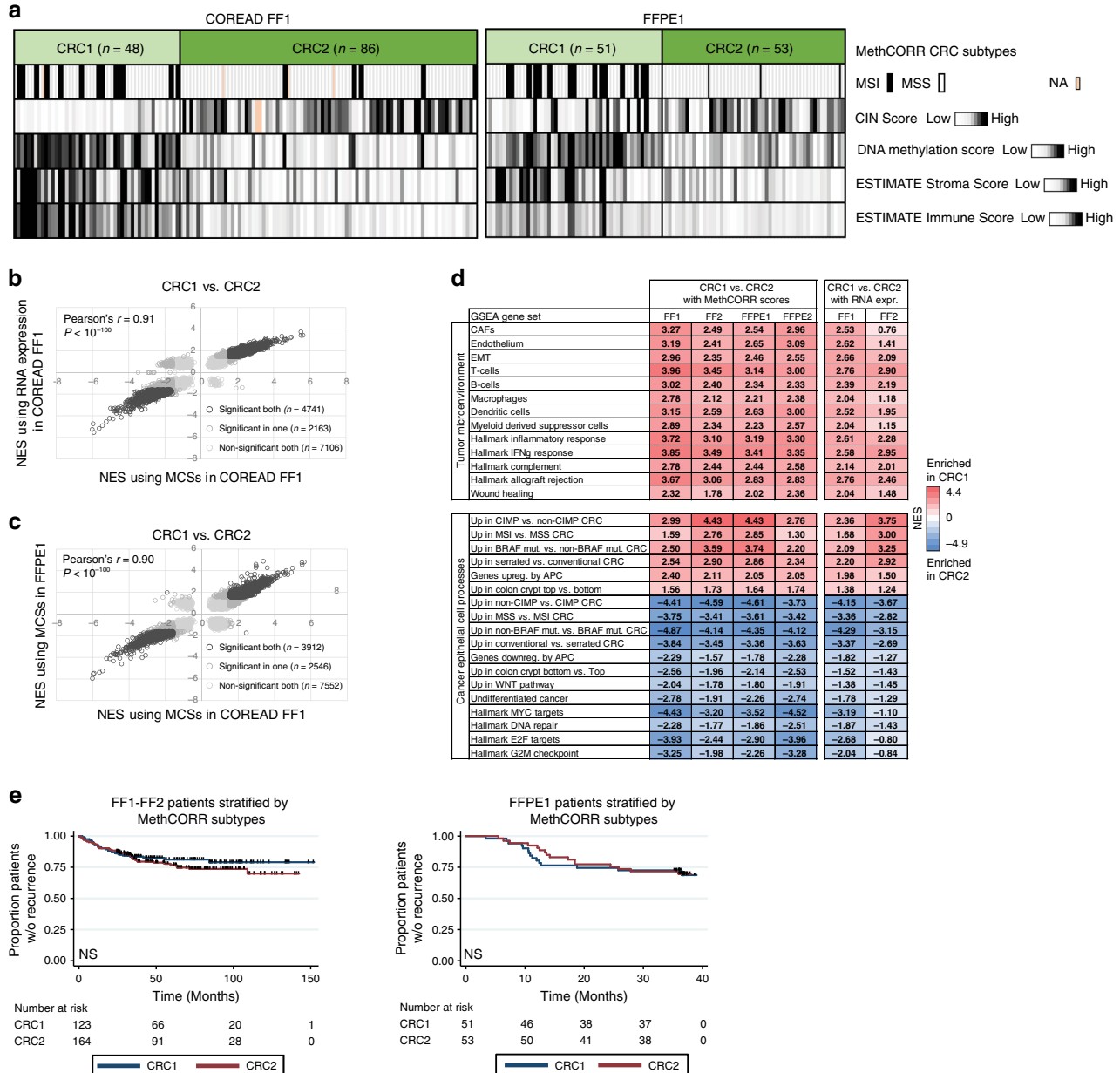

**Fig. 2 MethCORR based NMF clustering identifies the same two CRC subtypes in fresh-frozen and FFPE cohorts. a** Main molecular features of the CRC1 and CRC2 MethCORR subtypes in the COREAD FF1 and the FFPE1 cohort (Supplementary Table 1). MSI and MSS status is indicated in black and white. CIN scores were derived for COREAD and FFPE1 samples using GISTIC and EPIC DNA methylation data, respectively, and sample DNA methylation scores were calculated as the 40th percentile of DNA methylation β-values for all CpGs. Stroma- and Immune Scores were generated from MCSs using the ESTIMATE software[69]. **b–c** Scatterplots showing the correlation between normalized enrichment scores (NESs) for ~17 K gene sets of The Molecular Signatures Database (MSigDB) v6.1 from a pre-ranked GSEA of CRC1 vs. CRC2 subtypes in the COREAD FF1 cohort using either MCSs (*X*-axis) or RNA expression (*Y*-axis) as input (**b**) and a pre-ranked GSEA of CRC1 vs. CRC2 subtypes in either the COREAD FF1 cohort (*X*-axis) or FFPE1 cohort (*Y*-axis) using MCSs as input (**c**). Pearson's *r* and *P* value (Wilcoxon rank sum test) is indicated. **d** Table showing selected gene sets differentially enriched between CRC1 and CRC2 subtypes as evaluated by pre-ranked GSEA performed using MCSs or RNA expression in the fresh-frozen COREAD FF1 and SYSCOL FF2 cohorts and MCSs for the FFPE cohorts (Supplementary Table 1). Gene sets with positive NES are enriched in CRC1 (red colors), whereas negative NES indicate enrichment in CRC2 (blue colors). Gene sets enriched/depleted at a high significance level are highlighted in bold (FDR < 0.05). See methods section and Supplementary Table 3 for origin of gene sets. **e** Kaplan–Meier plots showing the relapse-free survival of CRC patients stratified according to subtype. Left panel: patients with fresh-frozen tumors and good clinical follow-up (the COREAD FF1 and SYSCOL FF2 cohorts; Supplementary Table 1) were combined to increase the number of relapse events. Right panel: patients with FFPE tumors and good clinical follow-up (The FFPE1 cohort; Supplementary Table 1). Significance was evaluated by the log-rank test.

that most CpGs were not located on the same chromosome as the gene they correlate with (Supplementary Fig. 4a). Instead, the most frequently occurring CpGs were located in genomic regions that exhibited great cell-type-specific variation in DNA methylation, as evaluated in 17 tissue types (GSE50192[18]; Supplementary Fig. 4b). Hence, the MethCORR matrix may help associate gene expression with particular cell types by comparing the methylation pattern of expression-correlated CpGs to known DNA methylation (or DNAse I hypersensitivity) profiles of cell monocultures/homogenous cell preparations. Indeed, expression-correlated CpGs for the T-cell-specific *CD3 Epsilon* (*CD3E*) gene overlapped with T-cell specific DNase I hypersensitive sites and DNA methylation patterns characteristic of T-cells (Supplementary Fig. 4c, d). Similarly, expression-correlated CpGs for *fibroblast activation protein alpha* (*FAP*) and *epithelial cellular adhesion molecule* (*EPCAM*) overlapped with patterns characteristic of stromal cells/fibroblasts and intestinal epithelial cells, respectively (Supplementary Fig. 4c, d). We also found that the genes with greatest expression-correlated CpG site overlap with *CD3E*, *FAP*, and *EPCAM* were themselves significantly associated with T-, stromal/fibroblast-, and epithelial-cell activities as evaluated by gene list enrichment analysis[30] (Supplementary Fig. 4e; *P* value < 0.05 by the Enrichr software[30]). This showed that analysis of expression-correlated CpGs help identify clusters of co-expressed genes and link them to particular cell types via comparison to cell-type-specific DNA methylation profiles.

To analyze expression correlations in a genome-wide format, we created a MethCORR map by clustering all MethCORR genes according to their overlap in expression-correlated CpGs (Fig. 3a). Foremost, the map was used to visualize differences between CRC1 and CRC2 by coloring gene nodes according to their difference in median MCS *z*-score between the subtypes (Δmedian *z*-score; Fig. 3a). The differences were near-identical for FF and FFPE cohorts (Fig. 3a, b and Supplementary Fig. 5a; Δmedian *z*-score Pearson's *r* range: 0.88–0.97, *P* value < $10^{-100}$, WRS test) and near-identical to a MethCORR map comparing serrated/MSI and conventional adenocarcinomas from the 450K methylation dataset GSE68060[31] (Fig. 3c; Δmedian *z*-score Pearson's *r* range: 0.87–94, *P* value < $10^{-100}$, WRS test). Similar results were obtained when the map was overlain with MethCORR interpretation of a transcriptional gene set defining serrated vs. conventional CRC (Supplementary Fig. 5b; Pearson's *r* range = 0.94–98, *P* value < $10^{-100}$, WRS test; for comparison to MSI status, CIMP status, CMS- and CRIS-classification status see Supplementary Fig. 5c, d). This suggested that CRC1 and CRC2 subtypes resembles serrated/MSI and conventional carcinomas, respectively. In support, Submap analysis confirmed that CRC1 and CRC2 subtypes from all four cohorts corresponded to the serrated/MSI and conventional subtypes from the GSE68060 dataset[31] (Supplementary Fig. 3d). Furthermore, CRC2 encompassed several map regions associated with high CIN scores, whereas CRC1 encompassed a large tumor microenvironment (TME) cluster characterized by genes with high stroma scores, as expected for conventional and serrated/MSI tumor subtypes[2,32], respectively (Fig. 3d).

**The MethCORR map characterizes intra-tumor heterogeneity.** To investigate the large TME cluster in greater detail and provide insight into sources of CRC heterogeneity, the map was overlain with MCS *z*-scores calculated from DNA methylation profiles of epithelial, immune, stem, and mesenchymal cells (primarily cell monocultures; Supplementary Table 4 and Supplementary Data 13). This identified map regions representing CAFs, CD14+ monocytes, CD3+ T cells, and CD19+ B cells among others (Fig. 3e). Again, similar results were obtained when the map was

overlain with MethCORR interpretations of RNA-based bio-markers and signatures defining CAFs, endothelium, myeloid cells, T cells, and B cells (Supplementary Fig. 5e). Hence, the MethCORR map can suggest cell types associated with RNA biomarkers and signatures via comparison to known cell-type-specific methylation profiles.

Based on this, we envisioned that the MethCORR map would visualize and suggest sources of inter-tumor heterogeneity between and within subtypes. CRC heterogeneity can arise from both differences in TME cell composition and in the differentiation status of tumor epithelial cells. For example compared with normal mucosa, CRCs can lose mature enterocyte traits and rather resemble enterocyte precursors, transit amplifying (TA) and stem cells, or undergo EMT[2,33,34]. Mapping of MCS *z*-scores from individual tumors revealed inter-tumor heterogeneity in both subtypes. For CRC1, heterogeneity was pronounced in the TME cluster and few samples had a dominant epithelial pattern (Fig. 3f). Three TME patterns were frequently observed, one overlapping with CAF/fibroblast (CAF/fibroblast pattern), another with CD14+ monocytic cells/platelets (inflammation pattern), and the last with lymphocytic T cells and B cells (lymphocyte pattern; Fig. 3e–g). This suggested that TME cell composition is a major contributor to intra-subtype heterogeneity in the immune-infiltrated CRC1 subtype. The TME patterns were less dominant among CRC2 samples (Fig. 3h) consistent with CRC2 conventional-like tumors being less immune-infiltrated[2] (Fig. 2a, d). Instead, CRC2 heterogeneity was pronounced within epithelial map regions and four patterns were observed (Fig. 3h): Two regions were dominated by signatures of enterocyte precursors and TA cells as estimated by overlapping the map with RNA signatures defining specific differentiation states of intestinal epithelial cells[33] (Fig. 3i). A third region overlapped with a mature enterocyte signature characteristic of normal mucosa samples (Fig. 3i and Supplementary Fig. 5f). Finally, an EMT pattern was identified in CRC2 by overlaying the map with MCSs of Hela cells undergoing EMT[35] (Fig. 3i) and GSEA showed enrichment of EMT signatures in the CRC2 samples with this EMT pattern (as compared with an early enterocyte pattern; Supplementary Fig. 5g). Collectively, this suggested that epithelial differentiation status is an important contributor to heterogeneity in the CRC2 subtype. Finally, the above heterogeneity was also identifiable among CRC cell lines and CMS subtypes (Supplementary Fig. 5h, i).

**MethCORR interprets prognostic RNA signatures.** We next investigated if MethCORR would also help identify DNA methylation-based biomarkers suited for prognostication using FF and FFPE samples. Our strategy was to use the MethCORR map to interpret established, prognostic RNA signatures and suggest cell types associated with tumor aggressiveness, which can be evaluated in DNA samples based on the cell-type specificity of methylation. Analysis of five prognostic signatures, CRC-113[36], ColoGuideEx[37], Oncotype DX[38], ColoPrint[39], and Tian et al.[40] showed that MCSs for almost all stromal transcripts were positively correlated with the median MCS for all signatures (Fig. 4a). This suggested that all signatures associated high TME activity with poor prognosis. MethCORR map analysis of the signatures revealed two distinct patterns within the TME cluster: The CRC-113, ColoGuideEX, and Oncotype DX signatures associated with a CAF-like pattern (Figs. 3e, f, and 4b), cancer invasiveness and *hepatocyte growth factor (HGF)* expression[41] (Fig. 4c, d). The ColoPrint and Tian et al. signatures (Fig. 4e) associated with an inflammation/wound healing pattern (Figs. 3e, f, and 4c) encompassing blood platelets, CD14+ monocytes (Fig. 3e), and *transforming growth factor beta 1 (TGFB1)* expression (Fig. 4d).

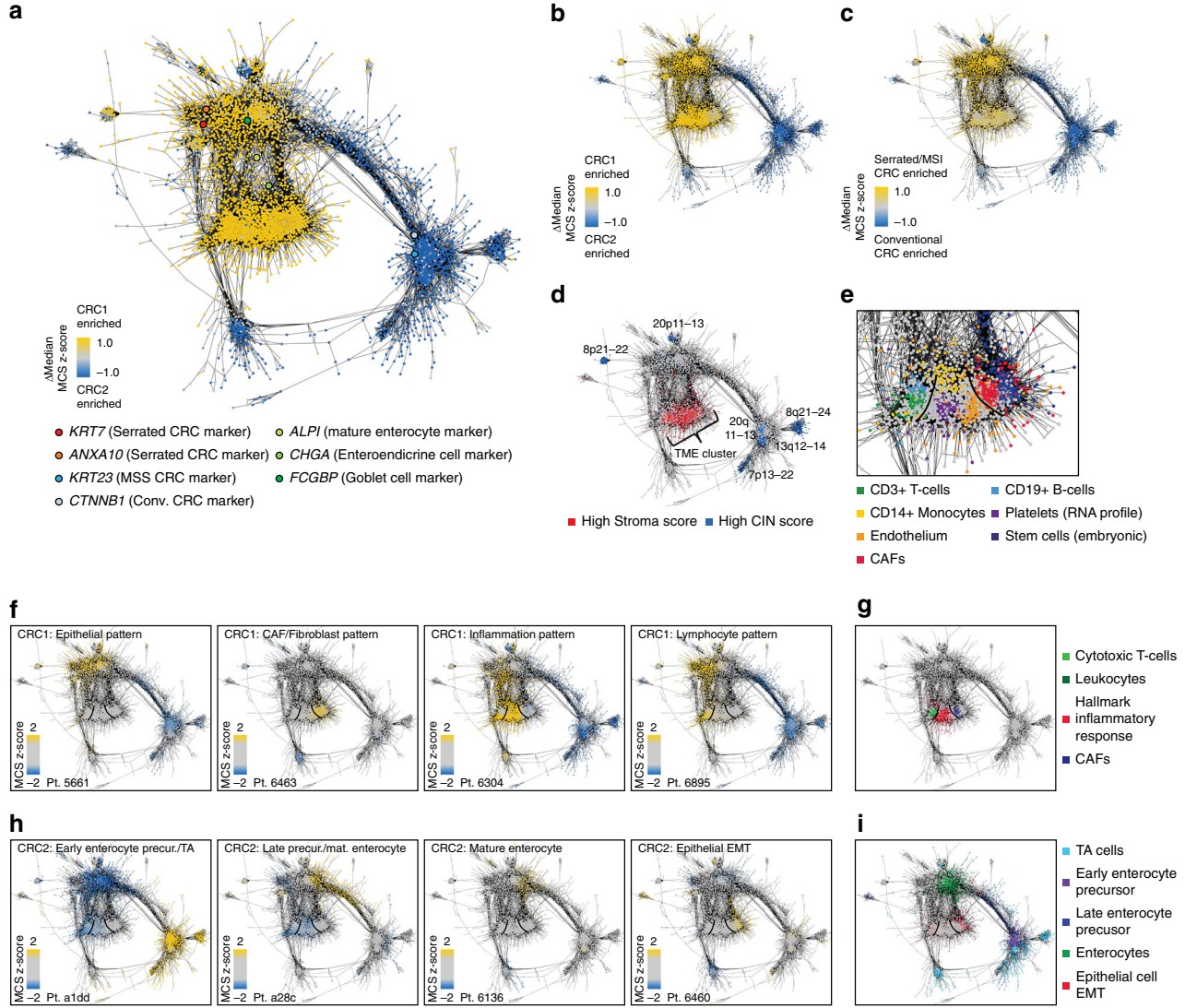

**Fig. 3 A MethCORR map identifies characteristics of CRC subtypes and intra-subtype heterogeneity. a** The MethCORR map is a representation of the MethCORR matrix established by clustering genes (cluster nodes) according to their overlap in expression-correlated CpGs (cluster edges) using Enrichment Map[63]. Each gene is colored according to the difference in median MCS z-scores (ΔMedian MCS z-score) comparing CRC1 and CRC2 within the COREAD FF1 cohort (Supplementary Table 1). Epithelial and CRC-related genes are highlighted by circles. **b** MethCORR map with genes colored according to ΔMedian MCS z-scores comparing CRC1 and CRC2 within the FFPE1 cohort. **c** MethCORR map with genes colored according to ΔMedian MCS z-scores comparing serrated/MSI and conventional CRCs (GSE68060[31]). **d** MethCORR map with genes colored according to a high stroma score (≥0.5[48]; red) or high CIN score (≥0.4; blue). A cluster encompassing genes with high stroma scores was named tumor microenvironment (TME) cluster. **e** Magnification of the TME cluster with genes colored according to high MCS z-scores for either CD3+ T cells, CD19+ B cells, CD14+ monocytes, platelets (RNA profile; MSigDB M7732), endothelium, stem cells (embryonic), or CAFs. MCS z-score profiles were calculated within a set of public DNA methylation profiles of cell monocultures and tissues (Supplementary Table 4 and Supplementary Data 13). Black lines indicate separation of the TME into lymphocyte, inflammation, and CAF/stem cell regions based on differences in cell-type composition. **f** and **h** MethCORR maps with genes colored according to the MCS z-scores of representative CRC1 (**f**) and CRC2 (**h**) samples calculated within all samples of the COREAD FF1 cohort. Black lines indicate TME patterns. **g** MethCORR map with genes colored according to high correlation to median MCS (cMCS) for three gene sets defining either cytotoxic T cells (MSigDB M13247[29] (BioCarta)), leukocytes[48], hallmark inflammatory response[29], or CAFs[48]. **i** MethCORR map with genes colored according to high cMCS for transcriptional gene sets up in either transit amplifying (TA) cells, early enterocyte precursors, late enterocyte precursors, or enterocytes[33]. Genes with >5% increase in MCSs during EMT of epithelial HeLa cells[35] are indicated in red. See methods section for details of ΔMedian MCS z-score and cMCS calculations.

Hence, the prognostic signatures overlapped in predictions, and pointed to CAF or inflammation/wound healing as associated with poor prognosis CRC. We recently reported that subtype-specific RNA signatures can improve prognostication beyond TNM staging in multiple CRC cohorts[3]. Therefore, MethCORR was also used to interpret these subtype-specific prognostic signatures denoted SSC prognosis and CIN prognosis. These are intended for immune-infiltrated/serrated and conventional carcinoma subtypes[3], which correspond to CRC1 and CRC2 in this study, respectively. MethCORR map analysis suggested that depletion of immune cells, including T cells, was associated with the SSC prognosis signature (Figs. 3e and 4c, f), whereas a CAF

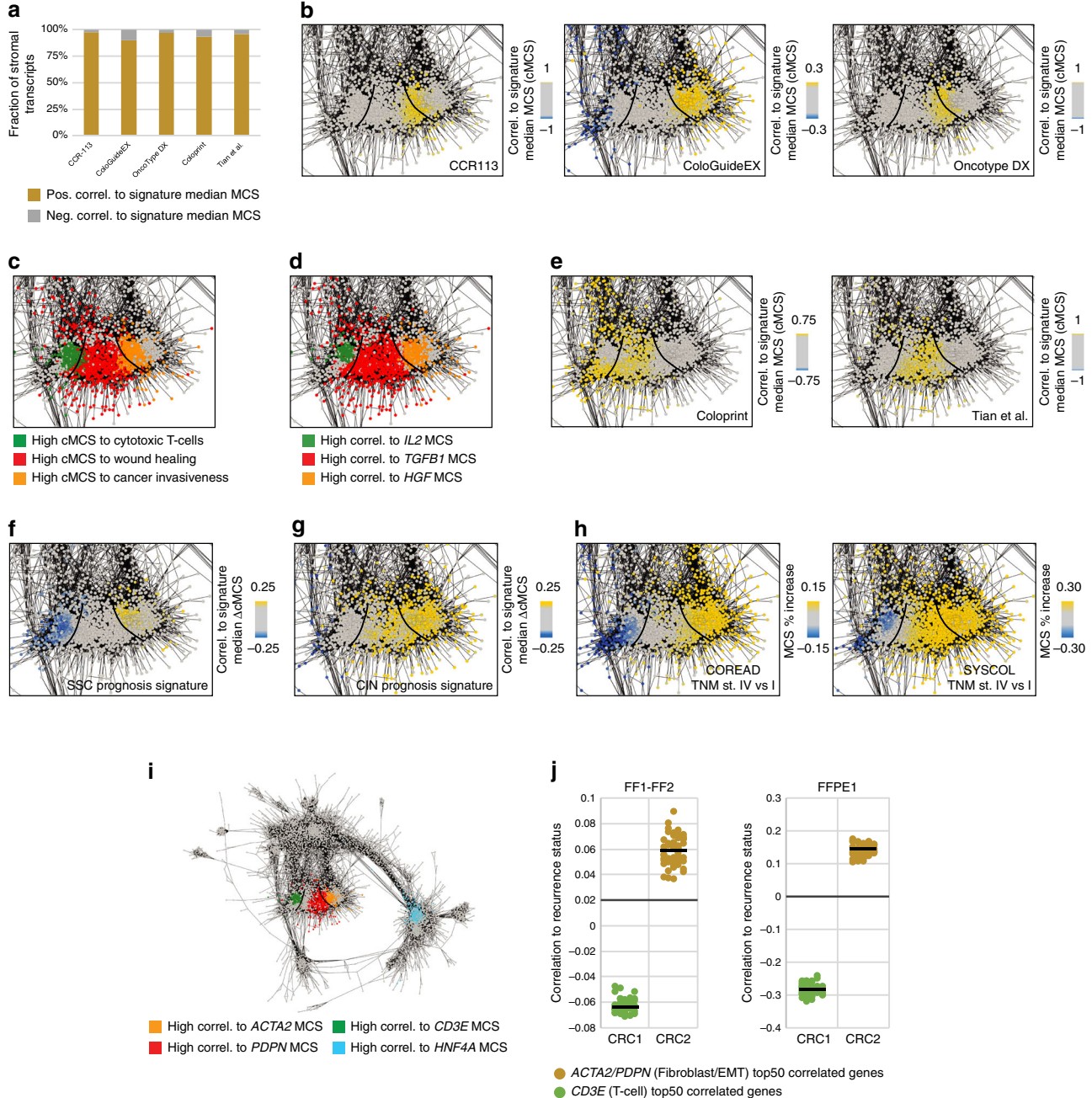

**Fig. 4 The MethCORR map suggests cell types associated with prognostic RNA signatures. a** Bar plot showing the fraction of stromal genes (stroma score > 0.5) that have positive or negative cMCSs (correlation to the median MCS) calculated for five prognostic RNA signatures CRC-113[36], CologuideEx[37], Oncotype DX[38], Coloprint[39], and Tian et al.[40] (see "Methods" for calculation of cMCSs). Stromal transcripts were significantly enriched among positively vs. negatively correlated transcripts for all five signatures ($P$ value < $10^{-100}$, Wilcoxon rank sum test). **b** Magnification of the TME cluster, where genes with the highest cMCSs for the prognostic CRC-113[36] (left), CologuideEx[37] (middle), and Oncotype DX[38] (right) signatures are highlighted. **c** Magnification of the TME cluster, where genes with the highest cMCSs for published gene sets defining cytotoxic T-cells (MSigDB M13247[29] (BioCarta); green), wound healing (MSigDB M11957[29,73]; red), or cancer invasiveness (MSigDB M2572[29,74]; orange) are highlighted. **d** Magnification of the TME cluster, where genes with the highest correlation to the MCS of the *IL2* (green), *TGFB1* (red), and *HGF* (orange) genes are highlighted. **e** Magnification of the TME cluster, where genes with the highest cMCSs for the prognostic Coloprint[39] (right) and Tian et al.[40] (left) signatures are highlighted. **f** Magnification of the TME cluster, where genes with the highest ΔcMCSs for the prognostic SSC prognosis signature[3] and **g** the CIN prognosis signatures[3] are highlighted. **h** Magnification of the TME cluster colored according to the gene-specific percentage change in median MCSs between TNM stage I and IV CRCs of the COREAD cohort (left) and SYSCOL cohort (right). **i** Magnification of the TME cluster, where genes with the highest correlation to the MCS of *CD3E* (green), *PDPN* (red), *ACTA2* (orange), and *HNF4A* (blue) are highlighted. **j** Scatterplot showing the Spearman rho for top *CD3E* or *ACTA2/PDPN*-correlated genes to positive relapse recurrence status in the CRC1 and CRC2 subtypes, respectively, in the fresh-frozen FF1–FF2 cohort (left) and FFPE1 (right) cohort. Median correlation is indicated by a black bar.

and EMT pattern was associated with the CIN prognosis signature (Figs. 3e and 4c, g). Furthermore, we compared MCSs for TNM stage I (favorable prognosis) to stage IV tumors (poor prognosis) in the COREAD and SYSCOL cohorts. Here, the relative change in MCSs between TNM stages also pointed to a relative loss of immune cells and increase in CAF content in late-stage, poor prognosis CRC (Fig. 4h). Collectively, the MethCORR analysis of seven published prognostic signatures hereby suggested that poor prognosis is associated with low T-cell content, particularly in the immune-infiltrated CRC1 subtype (Fig. 4f), or high CAF content and inflammation-EMT, particularly in the immune-depleted CRC2 subtype (Fig. 4g). To investigate the predictions of prognostic cell types in our FF and FFPE cohorts, we selected the three biomarkers *CD3E*, *ACTA2*, and *PDPN*. These are well-known markers for T cells[42], CAF/myofibroblasts[43], and inflammation-EMT[44], respectively, and their most closely CpG site-associated genes overlapped with regions highlighted by the prognostic classifiers (compare Fig. 4b, e, f, g, i; Supplementary Fig. 6). Indeed, top *CD3E*-associated genes negatively correlated with patient recurrence status in the CRC1 subtype and *ACTA2/PDPN*-associated genes positively correlated to patient recurrence in CRC2 (Fig. 4j).

**DNA methylation-based biomarkers for CRC prognostication**. To derive DNA methylation biomarkers for the above prognostic cell types we exploited the cell type-specificity of DNA methylation. Comprehensive comparison of multiple cell types identified low methylation of CpGs within the *CD3E*, *ACTA2*, and *PDPN* promoter as biomarkers for T cells, CAFs/myofibroblasts, and inflammation-EMT, respectively (Fig. 5a; Supplementary Data 13). Indeed, analysis of promoter CpGs in CRC samples showed that high methylation of the *CD3E* promoter, reflecting low levels of T-cell infiltration, associated with significantly poorer RFS in CRC1 in both FF and FFPE cohorts (Fig. 5b). In addition, low *ACTA2/PDPN* promoter methylation, reflecting high CAF/EMT levels, associated with poor RFS in CRC2 (Fig. 5b). The biomarkers were superior predictors of RFS as compared with TNM staging and MSI status (Fig. 5c, Supplementary Fig. 7a, b), and the biomarkers were only prognostic within the intended subtype (Supplementary Fig. 7c). Finally, to provide a cost-effective alternative to genome-wide methylome analysis, we evaluated *CD3E*, *ACTA2*, and *PDPN* promoter methylation using quantitative methylation-specific PCR (QMSP) assays. In addition, a QMSP assay targeting the *HNF4A* promoter was included for CRC subtyping; *HNF4A* is upregulated in CRC2 (Fig. 4i) and correspondingly, its promoter is less methylated in CRC2 (Fig. 5a). We applied our four biomarker assays to FFPE1 cohort samples, stratified patients into CRC1 and CRC2 using the *HNF4A* QMSP assay (Fig. 5d), and used *CD3E* and *ACTA2/PDPN* assays as prognostic biomarkers in CRC1 and CRC2. RFS analysis confirmed that the QMSP assays allowed subtype-specific prognostication using FFPE samples (Fig. 5e and Supplementary Fig. 7d).

**Discussion**
We here introduce MethCORR as an approach for uniform molecular analysis of FF and FFPE samples based on DNA methylation profiling. MethCORR allows inference of expression information from DNA methylation for a large number of genes (>11,000; Fig. 1). The inferred expression profiles support identical subtype discovery, characterization, and prognostication in FF and FFPE cohorts (Figs. 2–5). Notably, MethCORR allows three layers of information to be extracted from a DNA methylation array experiment, namely an inferred gene expression profile, a DNA methylation profile and a chromosome copy-

number profile, calculated from the methylation array signal intensity[45]. This improves cost-effectiveness and makes Meth-CORR attractive for analysis of archival FFPE material, where RNA profiling can be difficult[6–9]. The MethCORR concept bears resemblance to transcriptome-wide association studies, where gene expression is correlated to genetic variation. However, MethCORR allows the expression of many more genes to be modeled, which indicates that gene expression is stronger associated with DNA methylation than genetic variation[46,47].

The high number of MethCORR genes with inferred expression may be surprising, as several previous studies reported more infrequent correlations, when investigating associations between gene expression and methylation at local enhancers, promoters, and gene bodies[20–22]. MethCORR instead performs correlation analysis genome-wide and hereby identify far more associations from which expression information can be inferred. Indeed, expression-correlated CpGs were often located far from the gene locus, in regions with cell-type-specific methylation (Supplementary Fig. 4). Hence, MethCORR benefits from associating cell-type-specific gene expression with cell-type-specific DNA methylation patterns to infer expression information for many genes, even if associations are not functionally linked. Such indirect associations are expected in heterogenous cancer samples, which vary in their content of cancerous and non-cancerous cell types[2–4,48]. Support for a genome-wide correlation strategy is also found in two previous studies, which on a smaller scale, performed RNA expression-correlation analysis with more distantly located CpGs[49,50]. However, these studies only included ~500 CpG sites distributed across the genome compared with 480,000 sites utilized in MethCORR, and consequently found much fewer strong correlations.

MethCORR introduces an expression-correlated measure, the MCS, which enabled identification of the same two CRC subtypes in all four cohorts analyzed, and this independent of the analyzed tissue being FF or FFPE. The subtypes resemble the two major carcinogenesis pathways described in CRC[32] that are characterized by epithelial-cell hyper-methylation or chromosomal instability (Figs. 2 and 3). We speculate that MethCORR identified these well-established carcinogenesis pathways due to the relative emphasis of MCSs on cancer epithelial traits over stroma-related traits (Supplementary Fig. 3a, b). Also, we observed higher correlations between MCSs profiles for matched FF and FFPE biopsies taken from the same tumor than between RNA and iRNA profiles (Table 1). We therefore speculate that MCS-based characterization and subtyping is more independent of sample preservation type, which now require further testing.

MethCORR also introduces a map that visualizes genome-wide associations between gene expression and DNA methylation in CRC (Fig. 3). We envision that MethCORR map analysis may provide a framework for more detailed characterization of FF and archival FFPE samples than categorical subtyping alone, e.g., to reveal cellular sources of inter-tumor heterogeneity (Fig. 3). In particular, we illustrated that the MethCORR map can help identify cell types associated with RNA signatures (Figs. 3 and 4) and hereby help to derive DNA methylation-based biomarkers suitable for FFPE samples (Fig. 5). Our MethCORR map analysis of several prognostic RNA signatures (Fig. 4) showed that they all predicted cancer aggressiveness to be associated with cell types within the TME: In particular, a high CAF content, inflammation-associated EMT, and low T-cell content were associated with poor prognosis (Fig. 4). This agrees with clinically promising biomarkers such as the Immunoscore[42] and Tumor-Stroma Ratio[51]. Our analysis of CRC subtype-specific prognostic RNA signatures offered additional resolution: the T-cell content was primarily prognostic within the immune-infiltrated CRC1 subtype,

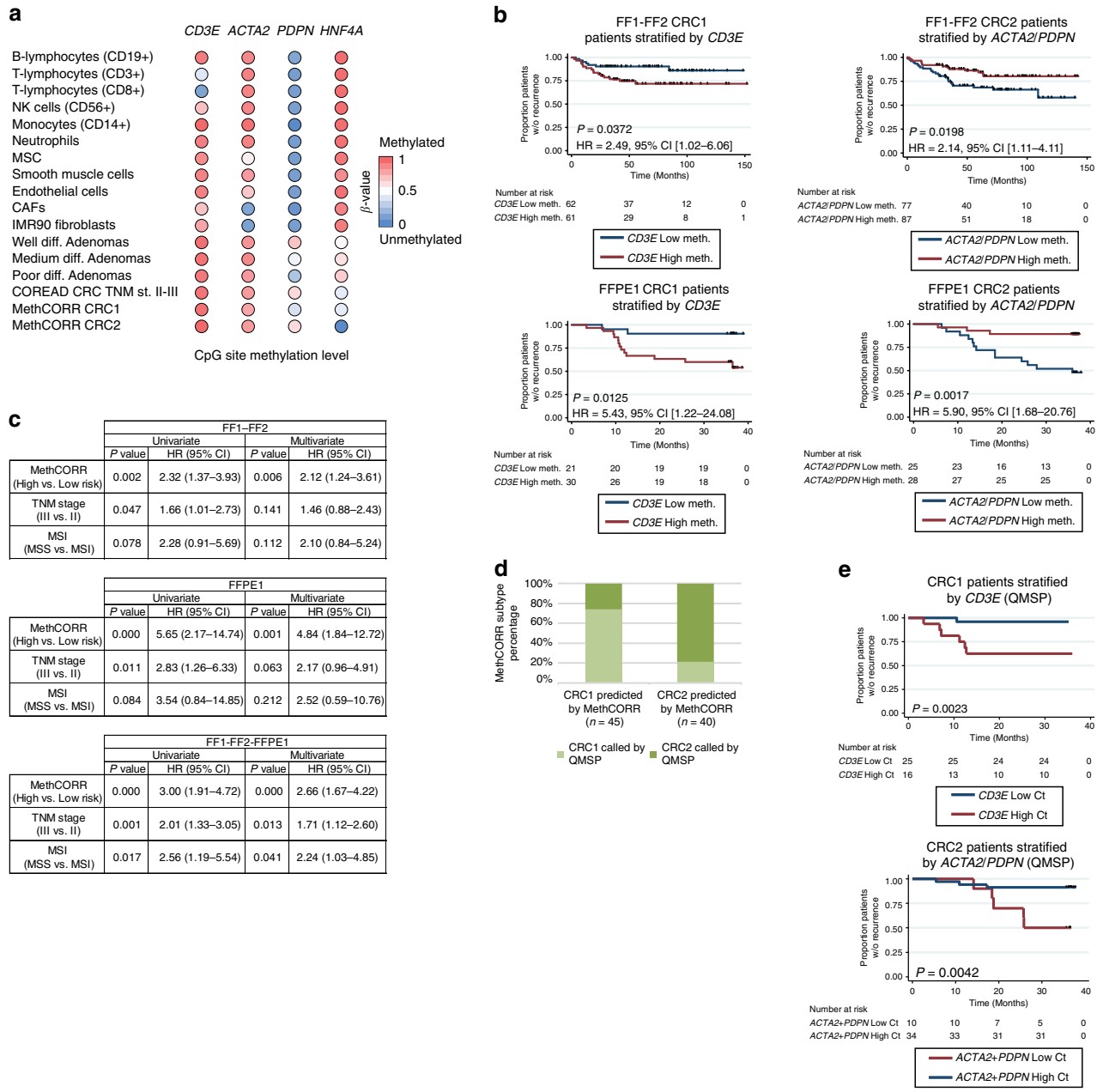

**Fig. 5 Validation of subtype-specific prognostic biomarkers in fresh-frozen and FFPE cohorts. a** Dot plot showing the methylation levels (β-values) of a CpG site in the promoter region of *CD3E*, *ACTA2*, *PDPN*, and *HNF4A* in selected cell types, adenomas and CRC samples as evaluated by the Infinium HumanMethylation450 BeadChip array. High and low methylation levels are indicated in red and blue colors, respectively. See Supplementary Data 13 for details of included cell types such as mesenchymal stromal/stem cells (MSCs), natural killer (NK) cells, and cancer-associated fibroblasts (CAFs). **b** Kaplan–Meier plot showing the relapse-free survival of patients stratified by the CpG methylation level of the *CD3E* promoter in CRC1 and by the average CpG methylation level of the *ACTA2/PDPN* promoter in CRC2 of the combined FF1-FF2- and the FFPE1 cohorts. *P* values (log-rank test) and HR95% CI are indicated. The same β-value cutoff was used in both cohorts (different cutoff for the subtype-specific biomarkers). **c** Table showing an uni- and multivariate cox regression analysis with MethCORR high and low relapse risk groups (a high relapse risk group was samples with high *CD3E* methylation levels in CRC1 or low average *ACTA2/PDPN* methylation levels in CRC2), TNM stage, and MSI status in the combined FF1–FF2 cohort, the FFPE1 cohort, and all cohorts combined. **d** Histogram showing the overlap in CRC1 and CRC2 status prediction by NMF clustering using MCSs or by QMSP in FFPE samples from the FFPE1 cohort. **e** Kaplan–Meier plot showing the relapse-free survival of CRC1 patients stratified by *CD3E* QMSP assay ΔCt values and in CRC2 by *ACTA2/PDPN* QMSP ΔCt-values in a total of 85 FFPE samples from the FFPE1 cohort. *P* values (log-rank test) and HR95% CI are indicated.

whereas CAF-content/inflammation-EMT was only prognostic in the less immune-infiltrated CRC2 subtype (Fig. 5). This supported our previous observations of subtype-specific prognostic biomarkers[3]. To aid further testing of subtype-specific prognostication, we established four simple QMSP assays for cost-efficient CRC subtyping and prognostication. The

application of the four QMSP assays in CRC samples confirmed and reproduced the RFS analysis derived from the more costly DNA methylome profiles (Fig. 5). Collectively, this illustrates the ability of MethCORR to help derive DNA methylation biomarkers from transcriptional signatures by extracting cell-type information from their expression-correlated CpGs.

Finally, MethCORR can provide high-quality gene expression measures in samples with poor RNA quality, such as archival FFPE samples for which confident RNA profiling is challenging[6–9]. Our analysis of matched FFPE and FF tissue showed that iRNA expression profiles from FFPE tissue resembled the RNA-sequencing profiles of the FF tissue better than RNA-sequencing profiles of the FFPE tissue. In PCA, matched FFPE iRNA and FF RNA-sequencing profiles clustered sample wise, while matched RNA-sequencing profiles of FFPE and FF tissue clustered according to preservation type. Preservation type-dependent clustering of FFPE and FF RNA-sequencing profiles have been reported previously, even in studies that report very high correlation between RNA-sequencing profiles of matched FFPE and FF samples[52,53]. We acknowledge that recent studies focusing on newly produced FFPE samples with optimal fixation and short storage time have reported improved correlations between matched FFPE and FF RNA-sequencing profiles[53–55]. However, such samples are not standard in the clinical FFPE archives. A large study, focusing on clinical FFPE samples, stored for many years, found that gene expression quantification was achieved in only 60% of samples and that correlation between biological replicates was very variable[8].

The robustness of MethCORR likely reflects that the Illumina Infinium HumanMethylation microarray produces highly concordant results in FFPE and FF samples when using DNA restoration for FFPE samples (Supplementary Fig. 2h)[14–17]. Furthermore, the DNA methylation $\beta$-values are calculated as the ratio between methylated and unmethylated CpG sites at a given genomic position. Hence, although a genomic region is affected by degradation, the ratio between the methylated and unmethylated fragments (i.e., the DNA methylation $\beta$-value) would expectably be robust. By contrast, RNA profiling is highly affected by RNA degradation[26] and the RNA quality obtainable from FFPE is often compromised[6–9]. In agreement, tumor samples with the lowest correlation between iRNA and measured RNA expression had lower RNA quality scores than samples with high correlations, whereas 450K methylation data quality did not differ (Supplementary Fig. 2g). This suggest that expression profiling of FF samples is influenced by even slight RNA degradation, as reported previously[26].

In conclusion, DNA methylation profiling and MethCORR analysis enables reliable and robust gene expression estimates to be obtained from clinical samples with compromised RNA quality. Furthermore, MethCORR data can be used to obtain clinically relevant information on tumor subtypes, cellular heterogeneity, and to develop prognostic biomarkers. Consequently, MethCORR represents an effective mean to unlock the unique and extensive resource of FFPE tissues in the pathology archives. We envision that MethCORR in the future will be established for many other cancer types.

## Methods

**CRC patient cohorts.** The COREAD cohort encompasses mucosa and UICC TNM stage I–IV CRC samples collected as part of TCGA project. All information regarding COREAD samples including processed DNA methylation data, RNA expression data, gene-level copy-number data, and clinical patient information (phenotype) were acquired via the UCSC XENA Public Data Hubs[24] [https://xena.ucsc.edu/public-hubs/] and the GDC Data Portal[25] [https://portal.gdc.cancer.gov/].

The SYSCOL and FFPE1 cohorts were acquired from the CRC biobank at the Department of Molecular Medicine, Aarhus University Hospital, Denmark. SYSCOL samples were collected at hospitals in the central region of Jutland, Denmark from 1999–2013[3]. The FFPE1 cohort encompasses CRC samples from the prospective study COLOFOL[56] collected at hospitals in the central region of Jutland, Denmark. None of the patients received neoadjuvant therapy. The tumors were histologically classified and staged according to the UICC TNM staging system. Cancer cell percentage was evaluated individually by two trained researchers, and when necessary, tumor biopsies were macroscopically trimmed to enrich the fraction of neoplastic cells. The SYSCOL and COLOFOL study was conducted in accordance with Danish law and is approved by local institutional

review boards and ethical committees and written informed consent was obtained from all patients. The FFPE2 cohort (IDIBELL) encompasses 56 samples collected at Medical Oncology Service of ICO Badalona-Germans Trias i Pujol Research Institute (IGTP), Spain. None of the patients received neoadjuvant therapy. The tumors were histologically classified and staged according to the UICC TNM staging system. Cancer cell percentage was evaluated individually by two trained researchers, and when necessary, tumor biopsies were macroscopically trimmed to enrich the fraction of neoplastic cells. Patients were followed according to the national clinical guidelines and written informed consent was obtained from all patients. Clinical information regarding the COREAD, SYSCOL, COLOFOL, and IDIBELL cohort samples is presented in Supplementary Table 1.

**DNA methylome data.** FF tumors from the SYSCOL cohort were macrodissected to enrich the fraction of neoplastic cells and DNA was extracted from serial cryosections using the Puregene DNA purification kit (Gentra Systems). Integrity of the genomic DNA from FF samples was assessed by 1.3% agarose gel analysis and only samples containing a high molecular weight smear (~50 KDa) were analyzed further. Bisulfite (BS) conversion of 600 ng DNA of each sample was performed according to the manufacturer's recommendations for the Illumina Infinium Assay (EZ DNA methylation kit. Zymo Research. Cat. No. D5004). Next, DNA methylation profiling was performed using Infinium HumanMethylation450 BeadChip technology (HM-450K; Illumina), as described by the manufacturer.

FFPE tumors from the COLOFOL FFPE1 cohort were macrodissected to enrich the fraction of neoplastic cells, DNA was extracted using the QIAamp DNA FFPE Tissue kit (Qiagen) and all samples passed the Infinium FFPE quality control (Infinium FFPE QC kit, Illumina). For methylation profiling 500 ng DNA underwent FFPE DNA restoration (Infinium HD FFPE DNA restore kit, Illumina) after BS conversion and profiling was performed using Infinium HumanMethylationEPIC BeadChip technology (HM-EPIC; Illumina), as described by the manufacturer.

FFPE tumors from the IDIBELL FFPE2 cohort were macrodissected to enrich the fraction of neoplastic cells. DNA was extracted using the QIAamp DNA FFPE Tissue kit (Qiagen) and all samples passed the Infinium FFPE quality control (Infinium FFPE QC kit, Illumina). For methylation profiling 250–500 ng DNA underwent FFPE DNA restoration (Infinium HD FFPE DNA restore kit, Illumina) after BS-conversion and profiling was performed using the Infinium HumanMethylation450 BeadChip technology (HM-450K; Illumina) as described by the manufacturer. For both the SYSCOL, FFPE1, and FFPE2 cohort the methylation $\beta$-values for each CpG site on the BeadChip were derived using the ChAMP R-package[57] using the champ.import and champ.norm functions. HM-450K DNA methylation profiles of the COREAD samples were acquired from the UCSC XENA Public Data Hubs[24] [https://xena.ucsc.edu/public-hubs/] and the GDC Data Portal[25] [https://portal.gdc.cancer.gov/] as normalized DNA methylation $\beta$-values. Missing $\beta$-values were imputed using the R-package Impute[58]. All DNA methylation measurements were performed once for each distinct sample.

**RNA-sequencing data.** FF tumors from the SYSCOL cohort were macrodissected to enrich the fraction of neoplastic cells and total RNA from serial cryosections were extracted using the RNeasy Mini Kit (Qiagen). RNA integrity was assessed using the Agilent RNA 6000 Nano Kit on an Agilent 2100 Bioanalyzer and >98% of analyzed samples had a RNA integrity number (RIN) > 6. Paired end mRNA sequencing was performed using 500 ng total RNA for library preparation with the TruSeq RNA Sample Prep Kit v2 and the TruSeq SBS Kit v3 was used for sequencing aiming for a minimum of 40 Million reads per sample. Sequencing reads were mapped to the human genome issue HG19 (hg19) using the Tophat2 mapper (Tophat: v2.0.10[59]) and estimating fragments per kilobase of exon per million fragments mapped (FPKM) values for Ensembl genes using Cufflink (Cufflinks: v2.2.1; Gencode v15 annotation w/o Pseudogenes[60]).

RNA-sequencing profiles for the COREAD samples were acquired from the UCSC XENA Public Data Hubs[24] [https://xena.ucsc.edu/public-hubs/] as $\log_2$(FPKM + 1) normalized RNA expression values for 20,530 genes and via the GDC Data Portal[25] [https://portal.gdc.cancer.gov/] as FPKM normalized RNA expression values for 60,483 transcripts. During comparison of RNA-sequencing data from nine matched FF and FFPE samples, only data originating from the same TCGA source center (indicated in Supplementary Data 11) were analyzed. Correlations between RNA sequencing in FF and iRNA expression in FFPE samples were analyzed using RNA-sequencing data from TCGA source center 22 (7 of 9 samples; 2 samples from TCGA source 23), as the GDC MethCORR matrix used for iRNA calculation was generated using RNA-sequencing data from samples primarily originating from TCGA source center 22 (76% of samples). All RNA-sequencing measurements were performed once for each distinct sample.

**Datasets used for MethCORR development.** The MethCORR development strategy was independently applied in three CRC datasets of paired RNA expression and DNA methylation data (Supplementary Data 1, 6, and 8) hereby generating three different MethCORR matrixes and sets of linear regression models. Primarily, MethCORR development was performed using Infinium Human-Methylation450K BeadChip (HM-450K) DNA methylation and RNA-sequencing

data from 394 samples of the COAD and READ cohorts (COREAD) of the TCGA project, acquired in normalized format via the UCSC XENA Public Data Hubs (Supplementary Data 1). The analysis was performed using $\log_2(\text{FPKM} + 1)$ normalized RNA expression values for all available 20,530 RNAs and DNA methylation $\beta$-values for the 396,065 CpGs, where $\beta$-values were provided by the XENA Public Data Hubs[24]. This analysis generated the COREAD MethCORR matrix (Supplementary Data 3) that is used for calculation of MCSs throughout the manuscript, unless otherwise indicated and modeling metrics is reported in Supplementary Data 2 and 4. Second, the MethCORR approach was applied to RNA-sequencing (20,336 RNAs) and HM-450K DNA methylation profiles (485,512 CpGs) from 314 samples of the SYSCOL cohort[3] (Supplementary Data 5–7) with the aim to validate the performance of the MethCORR approach in an independent cohort. Third, the MethCORR approach was applied to 405 TCGA COREAD samples using RNA expression (17,611 RNAs, these were selected from the original dataset of 60,483 transcripts as they overlap with the RNAs included in the UCSC XENA RNA dataset) and DNA methylation data (395,011 CpGs) acquired via the NCI GDC[25] (Supplementary Data 8). This analysis was performed to investigate the impact of RNA normalization methods on MethCORR performance (modeling metrics in Supplementary Data 9 and 10) and to generate a GDC data based MethCORR matrix that was used for analysis of the TCGA FFPE samples included in this study, as data from these FFPE samples were also acquired via the GDC database (Supplementary Data 11).

**Identification of RNA expression-correlated CpG sites**. The CRC cohort was divided in two discovery sets (sets 1–2, each encompassing 40% of samples), whereas a third set was reserved for independent validation (set 3, 20% of the samples; Fig. 1a and Supplementary Data 1, 6, and 8). Genome-wide correlations (Spearman) between the expression of each of the RNAs ($\log_2(\text{FPKM} + 1)$) and the DNA methylation $\beta$-value of each CpG site were calculated independently in discovery sets 1 and 2 using the publicly available R function "cor". All non-significant correlation pairs were discarded (Spearman's correlation $P$ value < 0.01). The remaining expression-correlated CpGs were ranked by their Spearman's rho in each discovery set and next by their rank sum within discovery sets 1 and 2 to identify top common expression-correlated CpGs. From these lists of ranked CpGs specific for each RNA, we selected up to 100 CpGs whose methylation $\beta$-value most negatively or positively correlated with its expression resulting in lists of ≤200 RNA expression-correlated CpGs for each RNA (depending on the number of expression-correlated CpGs in the ranked lists). To ensure analysis robustness, especially in FFPE samples, we excluded all CpG sites that had a detection $P$ value > 0.05 (ChAMP package[57]) in ≥5% of samples in either the SYSCOL, FFPE1, or FFPE2 cohort. Top ranking CpGs for all analyzed genes for the TCGA COREAD cohort (datasets acquired via the UCSC XENA Public Data Hubs) can be found in Supplementary Data 3.

**Calculation of MethCORR scores**. For each sample we used the methylation $\beta$-values of the top ≤200 RNA expression-correlated CpGs (for each gene) to calculate a MCS for all genes with both positively and negatively expression-correlated CpGs using the formula:

$$\text{MCS} = \frac{1}{\leq 200} \left( \sum_{1}^{\leq 100} \beta \text{ value pos. correl. CpG probe} + \sum_{1}^{\leq 100} 1 - \beta \text{ value neg. correl. CpG probe} \right).$$

The MCS formula calculates the average methylation value of the expression-correlated CpG sites specific for each gene. Unless otherwise indicated, the COREAD MethCORR matrix encompassing expression-correlated CpGs for 11,222 genes (Supplementary Data 3; MethCORR genes) was used for calculation of MCSs throughout the manuscript. The use of the MSC formula above and the MethCORR matrix provided in Supplementary Data 3 allow calculation of MCSs from DNA methylation $\beta$-values of any relevant 450K CRC data set of choice.

**Modeling and inferring of RNA expression from MCSs**. We modelled the relationship between MCSs and RNA expression for each gene in the discovery samples (set 1 + 2; Fig. 1A) using both simple linear ($\text{RNA} = B_0 + B_1 \times \text{MCS}$) and polynomial regression models ($\text{RNA} = B_0 + B_1 \times \text{MCS} + B_2 \times \text{MCS}^2 \ldots + B_n \times \text{MCS}^n$; $n = 2$–4). The Caret R-package[61] was used to perform modeling by $10 \times 10$-fold cross validation and we used the average RMSE to select the best model for each gene. As performances were highly similar for simple linear and polynomial models for most genes, we only selected polynomial models if a ≥5% relative decrease in RMSE values were observed over simple linear models. Model performances were independently validated in validation set 3 (Supplementary Data 2, 7, and 9). Genes with well-performing models ($R^2 > 0.16$ in both the discovery (set 1 + 2) and validation (set 3)) were regarded as MethCORR genes and included in the MethCORR matrix (Supplementary Data 3), whereas genes with poorer performing models were excluded. For MethCORR genes we inferred RNA (iRNA) expression for each gene in each sample using the MCS as input in the gene-specific linear regression models. Information of the gene-specific models are provided in Supplementary Data 2, which allow calculation of iRNA profiles from MCSs for any relevant 450K CRC data set of choice.

**Establishment and analysis of a MethCORR map**. The MethCORR map for the COREAD cohort was created by clustering MethCORR genes according to their overlap in expression-correlated CpGs using Cytoscape V3.2.0[62] and the application EnrichmentMap[63] (Jaccard + Overlap filtering cutoff 0.126). Only CpGs with negatively expression-correlated CpGs from the MethCORR matrix were used for identifying the overlap given that inclusion of all expression-correlated CpGs in a single map would complicate interpretation as genes with opposite expression-correlation to DNA methylation would cluster together. Genes with no significant CpG overlap to other genes are not included in the graphical representation of the MethCORR map for visual simplicity. For interpretation, the MethCORR map was overlain with several data types including external DNA methylation data, transcriptionally defined marker genes, gene sets, and signatures. To visualize these diverse data types using the MethCORR map, four types of scores were established as follows:

For DNA methylation datasets (450K/EPIC arrays), MCSs were first calculated for all samples and two types of scores were used for map visualization. The difference in median MCS z-scores (Δmedian MCS z-score) was used to visualize differences between subtypes encompassing multiple samples (such as between MethCORR subtypes, CMS subtypes, CRIS subtypes, MSI vs. MSS tumors etc.) whereas MCS z-scores were used for visualization of differences between individual samples within a cohort. MCS z-scores were calculated for each gene within each investigated cohort by subtracting the cohort mean from an individual sample MCS and dividing the difference by the cohort standard deviation. E.g. for analysis of inter-tumor heterogeneity, MCS z-scores were calculated for each gene within the whole COREAD FF1 cohort. For analysis of the cellular composition of the TME cluster, MCS z-scores were calculated from a collection of cell types with available 450K analysis downloaded from either Marmal-aid[64], Gene Expression Omnibus (GEO)[65], or Array express (see Supplementary Table 4 and Supplementary Data 13 for details of included samples; before calculation of MCS z-scores across all sample types the median MCSs were calculated for similar sample types, such as technical replicates).

For transcriptionally defined marker genes, gene sets, and signatures, two types of scores were used for map visualization depending on the data format. For simple gene sets and RNA signatures, defined by only one gene list (e.g., either up or downregulated RNAs), a correlation to median MCS (cMCS) was calculated for each MethCORR gene. The cMCSs were calculated as the average Pearson correlation between the median MCS of the gene set and the MCS of each MethCORR gene within the FF1, FF2, FFPE1, and FFPE2 cohorts. For complex gene sets/signatures, defined by two gene lists (e.g., of both up and downregulated genes), a correlation to median MCS difference score (ΔcMCS) was instead calculated for each MethCORR gene. The ΔcMCSs were calculated by subtracting the cMCSs for the downregulated gene set from the cMCSs for the upregulated gene set ($\Delta\text{cMCS} = \text{cMCS}_{\text{upreg.}} - \text{cMCS}_{\text{downreg.}}$) for each gene. For visualization, MethCORR map gene nodes were colored according to these MCS z-scores, ΔMCS z-scores, cMCS, and ΔcMCS as indicated in the text. For map visualization of published prognostic signatures, cMCS were calculated for the five general (non subtype-specific) signatures (CRC-113[36], ColoGuideEx[37], Oncotype DX[38], ColoPrint[39], and Tian et al.[40]), as they are single lists of RNAs associated with poor prognosis CRC (only recurrence score genes from the Oncotype DX panel were analyzed, whereas treatment genes were excluded). For the CRC subtype-specific SSC prognosis and CIN prognosis signatures ΔcMCS were calculated, as they are complex signatures encompassing lists of RNAs with high and low expression in aggressive CRC[3].

**NMF-based consensus clustering and SubMap analysis**. NMF consensus clustering was performed using the R-package NMF[66] with MCSs as input. The number of classes was determined by the first distinctive reduction in the cophenetic score and silhouette consensus score[67] and samples were classified according to consensus class. The similarity of independent subtype predictions was analyzed using the Genepattern SubMap module (v3[28,68]) using pairwise comparisons of MCSs and the following settings: num. marker genes = 50, number permutations for Fisher's statistics = 1000, weighted score type = no, null distribution = each. A false discovery rate (FDR) $P$ value < 0.05 was used as significance cutoff (provided by the Submap software[68]).

**CMS and CRIS subtype classification**. CMS classification was performed with the R-package CMSclassifier using the single sample method and nearest CMS as predicted subtype[2]. RNA expression or iRNA expression were used as input, as indicated in the text. CRIS classification was performed using the R-package CRISclassifier provided by Isella et al.[4] using RNA expression or iRNA expression as input, as indicated in the text.

**Stroma, CIN, DNA methylation, and ESTIMATE scores**. Stroma scores for each gene (fraction of reads of murine origin) was acquired from Isella et al.[48]. Genes with stroma scores >0.5 were considered stromal genes, whereas genes with stroma scores <0.1 were considered epithelial cancer genes. For the COREAD cohort, gene- and sample-specific CIN scores were established from the gene-level copy-number data (GISTIC2 analysis) available at the UCSC XENA Public Data Hubs[24]. The gene CIN scores were defined for each gene as the standard deviation of the

GISTIC2 copy-numbers of all samples within the COREAD cohort. The sample CIN scores were defined as the standard deviation of GISTIC copy-number scores across all genes within a sample calculated for each sample within the COREAD cohort. For non-COREAD cohort samples (without GISTIC2 data) CIN scores were derived from copy-number data extracted from the HM-450K/EPIC methylome BeadChips using the champ.CNA module of the ChAMP R-package[57]. Here, the sample CIN score was defined as the mean interquartile range of the copy-numbers for all chromosomal segments (seg.mean) covered by at least 25 Illumina probes (num. probes). The sample DNA methylation score for each sample was defined as the 40th percentile of DNA methylation $\beta$-values of all CpG sites common to all four CRC cohorts. ESTIMATE Stroma scores and Immune scores were calculated using the R-package ESTIMATE[69] using default parameters and MCSs as input. Household gene status was defined as genes included in the list of housekeeping genes[70] available at [https://www.tau.ac.il/~elieis/HKG/].

**Gene set enrichment analysis**. Pre-ranked GSEA was performed using the GSEA 3.0 tool[29] using default settings. Genes were pre-ranked according to their Spearman correlation of their MCS to CRC1 subtype status and gene sets were considered significantly up- or downregulated for FDR $q$ values < 0.05 (provided by the GSEA software[29]). The Molecular Signatures Database (MsigDB) gene set collection v6.1 was used with the addition of custom gene sets (Supplementary Table 3).

**Immunohistochemistry**. Immunohistochemical stainings of CRC tissue sections were acquired from the Human Protein Atlas[71] [https://www.proteinatlas.org/]. The following antibody and tissue sections were chosen (available from v8.proteinatlas.org): ACTA2 (antibody: CAB013531; Pt. 2001, Pt. 1898, Pt. 2468, Pt. 3074), PDPN (antibody: HPA007534; Pt. 2001, Pt. 1958, Pt. 1898, Pt. 3264), CD3E (antibody: HPA043955; Pt. 4724, Pt. 5005, Pt. 4448, Pt. 5004), HNF4A (antibody: CAB019417; Pt. 2001, Pt. 2151, Pt. 1958, Pt. 3074).

**Identification of cell-type-specific DNA methylation**. Genomic regions with cell-type-specific DNA methylation was identified by comparing multiple cell types with available 450K analysis downloaded from either Marmal-aid[64], GEO[65], or Array express[72]. Median MCSs were calculated for similar sample types, such as technical replicates prior to analysis (see Supplementary Data 13 for details). For selection of cell-type-specific methylation markers, only CpG probes with a ≥0.3 lower methylation $\beta$-value in the intended cell type, as compared with other relevant cell types, were selected. The following genes/CpG probes were included here: *CD3E*/cg24612198, *ACTA2*/cg09990481, *PDPN*/cg15563963, *HNF4A*/cg06640637.

**Quantitative methylation-specific PCR**. QMSP was performed using DNA primers specific for unmethylated *CD3E*, *ACTA2*, *PDPN*, and *HNF4A* gene promoter regions (See Supplementary Table 5 for primer sequences). BS conversion was performed with the EZ DNA Methylation-Direct™ Kit (ZYMO research) according to the manufacturer's protocol. QMSP was performed using the ViiA™ 7 Real-Time PCR system (Applied Biosystems). Biomarker assay reactions were carried out in triplicate in a final volume of 6 μl and contained 2.5 μl TaqMan® Universal PCR Master Mix, No AmpErase® UNG (Applied Biosystems), 0.15 μl of 20 pmol/μl forward and reverse primer, 0.2 μl of 5 pmol/μl hydrolysis probe, 0.125 μl TEMPase hot start DNA polymerase, 0.4 μl of 12.5 pmol/μl dNTP mix, 0.475 μl H2O, and 2 μl of 2.5 ng/μl BS treated DNA template. *AluC4A* reference gene reactions were carried out in triplicate in a final volume of 6 μl and contained 2.5 μl TaqMan® Universal PCR Master Mix, No AmpErase® UNG (Applied Biosystems), 0.2 μl of 25 pmol/μl forward and reverse primer, 0.1 μl of 17 pmol/μl hydrolysis probe, 0.125 μl TEMPase hot start DNA polymerase, 0.4 μl of 12.5 pmol/μl dNTP mix, 0.475 μl H2O, and 2 μl of 2.5 ng/μl BS treated DNA template. QMSP reactions were mixed in MicroAmp Optical 384 Well reaction Plates (Applied Biosystem) and run on the ViiA™ 7 Real-Time PCR system (Applied Biosystems) with the following PCR program: denaturation at 95 °C for 10 min, followed by 40 cycles at 95 °C for 15 s, and 60 °C for 1 min. The ViiA7™ software (Applied Biosystems) was used for evaluation of the fluorescence signals and the ΔCT was calculated by the use of the reference gene *AluC4A*. Subtyping was performed using ΔCT$_{HNF4A}$ as a marker for CRC2 and the average of ΔCT$_{CD3E}$ and ΔCT$_{ACTA2}$ as a marker for CRC1 (high stromal/immune cell infiltration). Samples with a ΔCT$_{HNF4A}$/(ΔCT$_{CD3E}$ + ΔCT$_{ACTA2}$)$_{average}$ ratio <0.85 were defined as CRC2 samples. CT values were measured three times for each sample (technical triplicates).

**Statistical analysis and RFS analysis**. Unless otherwise noted, statistical significance of differences between groups was determined using a non-parametric WRS test. During Submap analysis[28] and pre-ranked GSEA[29] a FDR-corrected $P$ value < 0.05 was considered significant and $P$ values were provided by the corresponding software. During gene list enrichment analysis an adjusted $P$ value < 0.05 was considered significant (provided by the Enrichr software[30]). During eFORGE analysis a Q-value < 0.05 was considered significant (provided by the eFORGE software). RFS analysis was performed in UICC TNM stage II–III samples with

good clinical annotation and follow-up (Supplementary Table 1). The inclusion criteria were as follow: A minimum of 2 years follow-up and survival after tumor resection, no local recurrence of the disease, no other cancer within 3 years, and no synchronous cancers. RFS was measured from date of surgery to verified first radiologic recurrence (distant) and was censored at the last follow-up or death. The following average normalized $\beta$-value cutoffs were used for the *CD3E*, *ACTA2*, and *PDPN* CpG probes to stratify patients into high- and low relapse risk groups: $\beta$-value$_{CD3E}$ < 1, average $\beta$-value$_{ACTA2, PDPN}$ ≤ 1. The following ΔCT cutoffs were used for the *CD3E*, *ACTA2*, and *PDPN* QMSP biomarker assays to stratify patients into high- and low relapse risk groups: ΔCT$_{CD3E}$ < 19.5, ΔCT$_{ACTA2}$ < 15.55, ΔCT$_{PDPN}$ < 13.5. Survival analysis was performed using the Kaplan–Meier method with the Stata/IC 14.2 (StataCorp) software. Significance is evaluated by log-rank test of equality. Cox proportional hazards regression analysis was used to assess the impact of MethCORR risk groups, TNM stage, and MSI status on RFS. The proportional hazard assumption was tested by a global test of the Schoenfeld residuals.

**Reporting summary**. Further information on research design is available in the Nature Research Reporting Summary linked to this article.

## Data availability
Normalized 450K DNA methylation datasets for the TCGA COREAD cohort used in this study are publicly available via the UCSC XENA Public Data Hubs[24] [https://xenabrowser.net/datapages/?dataset=TCGA.COADREAD.sampleMap%2FHumanMethylation450&host=https%3A%2F%2Ftcga.xenahubs.net&removeHub=http%3A%2F%2F127.0.0.1%3A7222] using the "dataset ID: TCGA.COADREAD.sampleMap/HumanMethylation450" and via the GDC Data Portal[25] [https://portal.gdc.cancer.gov/repository?facetTab=cases&filters=%7B%22op%22%3A%22and%22%2C%22content%22%3A%5B%7B%22op%22%3A%22in%22%2C%22content%22%3A%7B%22field%22%3A%22cases.project.program.name%22%2C%22value%22%3A%5B%22TCGA%22%5D%7D%7D%2C%7B%22op%22%3A%22in%22%2C%22content%22%3A%7B%22field%22%3A%22cases.project.project_id%22%2C%22value%22%3A%5B%22TCGA-COAD%22%2C%22TCGA-READ%22%5D%7D%7D%2C%7B%22op%22%3A%22in%22%2C%22content%22%3A%7B%22field%22%3A%22cases.samples.portions.is_ffpe%22%2C%22value%22%3A%5B%22false%22%5D%7D%7D%2C%7B%22op%22%3A%22in%22%2C%22content%22%3A%7B%22field%22%3A%22files.data_type%22%2C%22value%22%3A%5B%22Methylation%20Beta%20Value%22%5D%7D%7D%2C%7B%22op%22%3A%22in%22%2C%22content%22%3A%7B%22field%22%3A%22files.platform%22%2C%22value%22%3A%5B%22illumina%20human%20methylation%20450%22%5D%7D%7D%5D%7D] as "datatype=methylation beta value", "platform=illumina human methylation 450", and "case/biospecimen filter: samples portions is FFPE=false" for the TCGA-COAD and TCGA-READ project. Normalized RNA-sequencing data sets for the TCGA COREAD cohort used in this study are publicly available via the UCSC XENA Public Data Hubs[24] [https://xenabrowser.net/datapages/?dataset=TCGA.COADREAD.sampleMap%2FHiSeqV2&host=https%3A%2Ftcga.xenahubs.net&removeHub=http%3A%2F%2F127.0.0.1%3A7222] using the "dataset ID: TCGA.COADREAD.sampleMap/HiSeqV2" and via the GDC Data Portal[25] [https://portal.gdc.cancer.gov/repository?facetTab=cases&filters=%7B%22op%22%3A%22and%22%2C%22content%22%3A%5B%7B%22op%22%3A%22in%22%2C%22content%22%3A%7B%22field%22%3A%22cases.project.program.name%22%2C%22value%22%3A%5B%22TCGA%22%5D%7D%7D%2C%7B%22op%22%3A%22in%22%2C%22content%22%3A%7B%22field%22%3A%22cases.project.project_id%22%2C%22value%22%3A%5B%22TCGA-COAD%22%2C%22TCGA-READ%22%5D%7D%7D%2C%7B%22op%22%3A%22in%22%2C%22content%22%3A%7B%22field%22%3A%22cases.samples.portions.is_ffpe%22%2C%22value%22%3A%5B%22false%22%5D%7D%7D%2C%7B%22op%22%3A%22in%22%2C%22content%22%3A%7B%22field%22%3A%22files.analysis.workflow_type%22%2C%22value%22%3A%5B%22HTSeq%20-%20FPKM%22%5D%7D%7D%2C%7B%22op%22%3A%22in%22%2C%22content%22%3A%7B%22field%22%3A%22files.experimental_strategy%22%2C%22value%22%3A%5B%22RNA-Seq%22%5D%7D%7D%5D%7D] as "Experimental strategy=RNA-Seq", "Workflow Type= HTSeq = FPKM", and "case/biospecimen filter=samples portions is FFPE=false" for the TCGA-COAD and TCGA-READ project. 450K DNA methylation and RNA-sequencing data from TCGA CRC patient with matched FF and FFPE samples are publicly available via the GDC Data Portal[25] [https://portal.gdc.cancer.gov/] using the Database UUID provided in Supplementary Data 11. The RNA-sequencing data from the SYSCOL adenoma/carcinoma samples and the SYSCOL 450K methylome data is deposited at European Genome-phenome Archive (EGA, [https://www.ebi.ac.uk/ega/]), which is hosted by the European Bioinformatics Institute (EBI) and the Centre for Genomic Regulation (CRG). Study accession numbers are: EGAS00001002376 (RNA sequencing) and EGAS00001004293 (methylomes). The dataset and sample ID's of the other publicly available DNA methylation datasets used in this study are given in Supplementary Data 13. All other data supporting the findings of this study are available within the article, its supplementary information files and from the corresponding author upon reasonable request. A reporting summary for this article is available as a Supplementary Information file.

## Code availability
R codes for calculation of MCSs and iRNA profiles are available upon request.

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

## Acknowledgements

This research is supported by grants from the European Commission FP7 project SYSCOL (UE7-SYSCOL-258236), the Novo Nordisk Foundation (NNF16OC0023182), the Danish National Advanced Technology Foundation (056-2010-1), the John and Birthe Meyer Foundation, the Danish Council for Independent Research (Medical Sciences) (DFF − 0602-02128B, DFF – 4183-00619, DFF − 7016-00332B), the Danish Council for Strategic Research (1309-00006B), the Danish Cancer Society (R40-A1965_11_S2, R56-A3110-12-S2, R107-A7035, R133-A8520), the National Cancer Institute of the National Institutes of Health (R01 CA207467), the Aage and Johanne Louis-Hansen's Foundation (17-2-0457), Dansk Kræftforskningsfond (DKF-2017-26 - (26)), the Knud and Edith Eriksen's Memorial Foundation, the Neye Foundation, and the Manufacturer Einar Willumsen's Memorial Foundation (6000073). The Danish Cancer Biobank is acknowledged for biological material. We thank P. Celis, L. Nielsen, L. Kjeldsen, B. Devantie, B. Trolle, S. Moran, D. Garcia, and C. Arribas for their technical support. The results published here are in part based upon data generated by the TCGA Research Network [https://cancergenome.nih.gov/].

## Author contributions

T.B.M., C.L.A., and J.B.B. designed the experiments. T.B.M., M.H.R., J.S., H.O., S.S.A., J.G., A.M.C., M.C.M., A.H.M., S.L., E.T.D., M.E., C.L.A., and J.B.B. performed the experiments and included patients. T.B.M., M.H.R., C.L.A., and J.B.B. analyzed and interpreted the data. T.B.M., C.L.A., and J.B.B. drafted the manuscript. All authors reviewed and approved the final manuscript.

## Competing interests

The authors declare no competing interests.
