## [Peer Review File · Nature Communications]

Reviewers' comments:

Reviewer #1 (Remarks to the Author):

In their manuscript entitled "MethCORR: DNA Methylation-based Characterization, Classification and Prognostication of Colorectal Cancer using Archival Formalin-fixed, Paraffin-embedded Tissue", Bramsen and co-workers took advantage of publicly available datasets for which genome wide methylation and whole transcriptomic profiles were present.

Using correlation analyses between the two molecular layers, the authors identified a series of methylation probes that could be used as surrogate for specific gene expression in fresh frozen as well as in FFPE tissues. Such signature was also correlated with transcriptional subclassification and associated with specific prognostic value.

+ The "unsupervised" attempt to correlate gene transcription with epigenetic features wherever in the genome is of high interest and demonstrates the great ability of methylation in stratifying phenotypes. In fact methylation triggers changes in expression but will also be modified upon expression shift.

+ Since most of the recent classifications of colorectal cancer rely on transcriptomics which demonstrated big limitations with FFPE sample, the manuscript, which showed the ability of methylation to outperform transcription is timely and innovative.

+ The manuscript is also well written, but would benefit from a more divulgative methods paragraph for non statisticians / bioanalysts.

+ The authors should also make available in supplementary table the methylation probe / individual gene correlations as well as those probe coordinate used in the MethCORR map.

+ I wonder how other classifications correlate with the methCORR map.

In fact the authors demonstrated the ability conventional pathway versus serrated (MSI). What about CIMP classification? Hinoue and colleagues published in Genome Res 2013, the list of CpG probes from the 27 array (also used in the TCGA to determine the 4 classes of CIMP: CIMP-H/CIMP-L/CIMP-3/CIMP-4).

The authors mentioned the transcriptional CMS class, what about the CRIS defined by Isella et al in 2015?

+ How well would pre-clinical models (cell lines, patients' derived xenograft) perform using the MethCORR map?

Reviewer #2 (Remarks to the Author):

Mattesen and colleagues have undertaken an interesting project to understand colorectal cancer and to exploit methylation data as a way of bypassing FFPE challenges in RNA profiling. The idea is based on at least two assumptions:

- 1) that FFPE tissue is more amenable to methylation rather than RNA profiling (which is unclear)
- 2) that a transformation of methylation data into RNA space provides additional value (which is not shown)

Additionally, while this reviewer had not previously known of this strategy to infer RNA levels from methylation data, it is not novel. There are several publications doing it. Finally, several technical concerns with the actual modeling and validation procedures themselves were identified, and these are sufficiently large to make downstream interpretation of biological findings difficult.

1. FFPE RNA-Seq

The authors suggest that FFPE material is not amenable to high-quality RNA, and that this precludes robust profiling (lines 58-64). At this point, it is increasingly difficult to support this claim, as there are literally hundreds of papers reporting robust RNA-Seq on FFPE material. As quick examples, consider PMIDs: 29862382, 25495041, 31058252. And of course there are standard kits that claim to support such analyses: www.illumina.com/science/education/ffpe-sample-analysis/low-quality-ffpe-rna-seq.html

It simply isn't possible to support the idea that methylation profiling of FFPE material is significantly superior to RNA profiling of it. Both are possible, but both can fail in surprising ways, some of which appear to be a function of the specific fixation procedure and reagents, others of which are correlated to age, and still others of which are unknown. The authors use this as the motivation for the study, and this comprises a major weakness. In some sense, they are tackling a problem which is not current, and for which there is little evidence that methylation data will be universally superior to RNA data.

2. TWAS

The core of the method is learning a methylation-to-RNA model for each gene. This is directly analogous to a TWAS study, and it is surprising that the authors do not draw this comparison, nor outline the similarities and difference between the two.

3. Model validation

As was the case in many of the early TWAS studies, model validation here is insufficient. Considering the recent reviews and papers in that field would help significantly with the validation here. Key problems with the modeling and validation are:

- a) The models fail to adjust for critical variables like sex, ethnicity and age which are known to be methylation-associated
- b) The iRNA scores shown in Figure 1d do not appear to show uniform error, but instead to be associated with abundance level
- c) There is no gene-specific assessment of model fit
- d) Model that fit poorly appear to be included in downstream analyses
- e) Per-sample correlations are much less important than per-gene correlations across a cohort
- f) We are not shown the distributions of correlations across genes, making outlier analysis impossible
- g) All values are reported as Pearson's correlations, when there is no evidence to believe these relationships are linear (and indeed they clearly are not in some cases, e.g. Figure 1e shows definite non-linearity)
- h) Throughout Pearson's R is reported, but the meaningful metric here is R² (or some other metric of percent variance explained). The correlations of 0.8-0.9 are much less impressive in that space, with values 0.65-0.81, which means a third to a fifth of variance is not explained
- i) Beyond the choice of Pearson for correlations, and R vs. R² as a metric, a bigger problem is that correlations aren't the meaningful metric at all. Correlations are giving rank-order, but most RNA abundance analyses (indeed including the many done downstream of this work here) use continuous values not rank-ordered ones. As a result, residuals and similar metrics of deviation from truth are the key values. Those are not reported systematically, and we do not see their distributions
- j) There is little to no analysis of the features of consistently well- and consistently poorly-modeled genes
- k) Line 128, the authors attribute poor sample-wise correlation to low-quality RNA, but do not demonstrate whether these samples have equivalently low-quality methylation data.
- L) The core idea here is not novel, for example PMID:27897002, 29581450 did a simpler version of the modeling, while PMID:31106051 did a much more sophisticated version. Thus the core novelty of the methodologic approach is unclear.

a) The actual MCS is something like the average beta-value (scaled to give symmetry). It is unclear why this is reasonable, and why better-correlated probes are not given higher weights

Minor

* the critical MCS value is not defined in methods, but placed in a figure, which is a poor decision
(placing it in both is fine)

Reviewers' comments:

Reviewer #1 (Remarks to the Author):

Foremost we would like to thank reviewer 1 for the kind comments and inspiring suggestions. We have introduced several alterations to the manuscript to incorporate these suggestions.

In their manuscript entitled "MethCORR: DNA Methylation-based Characterization, Classification and Prognostication of Colorectal Cancer using Archival Formalin-fixed, Paraffin-embedded Tissue", Bramsen and co-workers took advantage of publicly available datasets for which genome wide methylation and whole transcriptomic profiles were present.

Using correlation analyses between the two molecular layers, the authors identified a series of methylation probes that could be used as surrogate for specific gene expression in fresh frozen as well as in FFPE tissues. Such signature was also correlated with transcriptional subclassification and associated with specific prognostic value.

+ The "unsupervised" attempt to correlate gene transcription with epigenetic features wherever in the genome is of high interest and demonstrates the great ability of methylation in stratifying phenotypes. In fact methylation triggers changes in expression but will also be modified upon expression shift.

Reply 1: we want to thank the reviewer 1 for the very kind comments. We are very happy that reviewer 1 sees the great potential in performing genome-wide correlation between RNA expression and DNA methylation to provide novel methylation-derived measures, here the MCSs, that can be used to characterize and stratify phenotypes.

+ Since most of the recent classifications of colorectal cancer rely on transcriptomics which demonstrated big limitations with FFPE sample, the manuscript, which showed the ability of methylation to outperform transcription is timely and innovative.

Reply 2: We want to thank the reviewer 1 for the very kind comments. We are very happy that the reviewer 1 finds our manuscript timely and innovative. In addition, we are encouraged that the reviewer acknowledges that transcriptomic analysis can have limitations in FFPE tissues, an observation that is well known to

researchers within the field but is more rarely represented in the published literature. In the revised manuscript, we have provided a more detailed analysis of RNA and DNA profiling in fresh-frozen and FFPE tissue (Figure 1g and supplementary 1o-q; also please see reply 13 and 17 below).

+ The manuscript is also well written, but would benefit from a more divulgative methods paragraph for non statisticians / bioanalysts.

Reply 3: We now provide a more elaborate description of the methods. Specifically, we have expanded Figure 1a and Supplementary Figure 1 to provide more details of MethCORR development. We have also rewritten and expanded the methods section describing the development of the MethCORR approach (see lines 489-586).

+ The authors should also make available in supplementary table the methylation probe / individual gene correlations as well as those probe coordinate used in the MethCORR map.

Reply 4: The COREAD MethCORR matrix (list of expression-correlated probes for each MethCORR gene) is now included it in Supplementary Table 3 to allow calculation of MethCORR scores and gene correlation in any independent cohort as well as to establish the MethCORR map.

+ I wonder how other classifications correlate with the MethCORR map.

Reply 5: We have included additional figures in Supplementary Figure 3j illustrating where the consensus molecular subtypes (CMS) and CRC intrinsic signature (CRIS) subtypes are located on the MethCORR map.

In fact the authors demonstrated the ability conventional pathway versus serrated (MSI). What about CIMP classification? Hinoue and colleagues published in Genome Res 2013, the list of CpG probes from the 27 array (also used in the TCGA to determine the 4 classes of CIMP: CIMP-H/CIMP-L/CIMP-3/CIMP-4).

Reply 6: We have now included an additional figure in Supplementary Figure 3i illustrating where the CIMP class are located on the MethCORR map. For this analysis, we used the subtypes provided by the TCGA COREAD clinical matrix acquired via UCSC XENA database comparing the annotated "PANCAN_DNAMethyl" classes "COADREAD non-CIMP c11" vs. "COADREAD CIMP c12". Unfortunately, too few sample annotations for the classes CIMP-H/CIMP-L/CIMP-3/CIMP-4 were provided in the downloaded XENA dataset for the COREAD FF1

samples to allow all four classes to be included in the map. Still, the CIMP class show great overlap with the MethCORR CRC1 subtype as is expected. Furthermore, a similar analysis for MSI vs MSS CRC is now included in Supplementary Figure 3i.

The authors mentioned the transcriptional CMS class, what about the CRIS defined by Isella et al in 2015?

Reply 7: We have included MethCORR maps showing the CMS and CRIS classes (see reply 5; Supplementary Figure 3j). We have also included a Supplementary Figure 1n that show the overlap between CRIS classification of TCGA COREAD samples using either RNA expression or imputed RNA scores (iRNAs) as input for classification using the CRIS approach (similarly to Figure 1f for CMS classes). Also for the CRIS subtypes, there is good concordance between the subtype calls obtained using RNA and iRNA data.

+ How well would pre-clinical models (cell lines, patients' derived xenograft) perform using the MethCORR map?

Reply 8: We have included comparison of CRC cell lines using the MethCORR map to demonstrate molecular heterogeneity within and between MSI and MSS cell lines (Supplementary Figure 3m). Considering the current length and complexity of the manuscript, we abstain from including evaluation of patient xenografts.

Reviewer #2 (Remarks to the Author):

Foremost, we would like to thank reviewer 2 for critically assessing our manuscript and for the many suggestions, which have guided the revision of our manuscript. We have re-written and re-calculated all results to incorporate these suggestions and hope that reviewer 2 would look favorably on our revised manuscript.

Mattesen and colleagues have undertaken an interesting project to understand colorectal cancer and to exploit methylation data as a way of bypassing FFPE challenges in RNA profiling.

The idea is based on at least two assumptions:

- 1) that FFPE tissue is more amenable to methylation rather than RNA profiling (which is unclear)

Reply 9: We fully agree that it is not well-studied if FFPE tissue is more amenable to DNA methylation than RNA profiling given that both DNA and RNA is modified by formalin fixation and degraded during long-term FFPE storage. Still, we believe that DNA methylation profiling by the Illumina Infinium HumanMethylation microarray technology provides a robust alternative to RNAseq for the following reasons:

1) DNA is, at least by some researchers, considered less sensitive to degradation than RNA in FFPE samples (PMID: 28246590) and a better source for robust biomarkers than RNA (PMID: 24937670). Also, In PMID: 28742856 the authors compares the fitness of nucleic acids from FFPE tissue extracts and concludes that “DNA derivatives to be easier to extract, more stable, and less prone to degradation than RNA”. In PMID 30192857, the authors compare the integrity of RNA and DNA in FFPE and cryo-preserved tissue by qPCR. They find a lower increase in CT values for DNA-based vs RNA-based qPCR assays (of similar length; 271-278 bp; compare figure 6 and 8) in FPPE as compared to frozen samples. The above notion has been mentioned in the revised manuscript in line 58-59)

2) In the revised manuscript we have now highlighted that enzymatic strategies for DNA repair/restoration have significantly improved the analysis of FFPE DNA for several types of analysis (e.g. PMID: 28209900), including DNA methylation profiling by Illumina Infinium HumanMethylation beadchip technology: Upon DNA restoration, the beadchip technology produces very high correlation between DNA methylation β -values in matched fresh-frozen and FFPE samples (e.g. PMID24732293: average Pearson R = 0.9721 (0.9258-0.9945);

PMID:30239063: average Pearson $R > 0.95$; PMID25867767; average Pearson $R = 0.9590$ (s.d. ± 0.0184); PMID: 28392843; average Pearson $R=0.938$ (std: 0.028)). Without DNA restoration such correlations are lower (PMID:25867767: Spearman correlation $\rho=0.8051$ (s.d. ± 0.1028) vs. $\rho=0.9590$ s.d. ± 0.0184); PMID:20434562: median $R^2 = 0.50$ (0.49–0.74) vs. $R^2 = 0.91$ (range 0.88–0.96)), especially when using low DNA input (PMID:30239063). The above notion has been mentioned in the revised manuscript in line 60-62 and 413-415. We are not aware that similar repair/restoration protocols exist for single-stranded, FFPE-derived RNA.

3) In the revised manuscript we now compare the correlation between RNA-sequencing profiles, or DNA methylation (HM-450K) profiles, of matched FF and FFPE samples from nine TCGA COREAD patients (Figure 1g and Supplementary Figure 1p) and find that correlations for DNA methylation data are significantly higher (median $R^2 = 0.96$, range 0.94–0.98) than for RNA seq (median $R^2 = 0.7$, range 0.63–0.87), $P < 0.001$ Wilcoxon rank sum test. In accordance, we now also show that our DNA methylation-based MCS profiles are also very robust between matched CRC FFPE and fresh-frozen samples (median intra-tumor correlation R^2 of 0.99, range 0.98–1; Figure 1g). We reached the same conclusion, that R^2 for MCS-MCS correlations are greater than RNA-RNA correlations in fresh-frozen and FFPE tissue, when we analyzed paired data from 26 patients from other cancer types (the BRCA, PRAD, LUAD, BLCA, KIRC, and UCEC from the TCGA dataset (MCS-MCS median $R^2 = 0.96$ (0.57–0.99) and RNA-RNA median $R^2 = 0.79$ (0.49–0.88), $P < 0.001$ Wilcoxon Rank Sum test). The data is part of a separate manuscript, and hence not included in the present manuscript.

4) The DNA methylation β -value of individual CpG sites is calculated as the ratio between the signal from methylated and unmethylated CpG DNA fragments (independently of other CpG sites). Hence, although a genomic region may be slightly degraded/inaccessible relative to other regions, the ratio between the methylated and unmethylated fragment within the region would be expected to give the same β -value independent of degradation. This notion has been mentioned in the revised manuscript in line 415–418. This is probably also the explanation why there is hardly any difference between β -values in comparisons between paired fresh frozen and fixed tissues (PMID: 24732293, PMID: 24878701, PMID: 29862382). In contrast, RNA quantification by RNA-seq is a relative measure and we used FPKM (Fragments Per Kilobase of transcript per Million mapped reads). Therefore, even small differences in RNA quality may shift the expression profiles of a sample due to RNA-species specific degradation (PMID: 24885439). Furthermore, NCI conducted a biospecimen preanalytical variable study showing that formalin fixation caused a “consistent and dramatic” shift in the proportion of intronic/exonic/untranslated RNA reads for most genes leading to a shift in expression signatures (PMID:31061401). Hence, for RNA sequencing, the observed inter-sample differences can be due to

both true biological differences, RNA species-specific differences in degradation and differences in RNA profiles introduced by formalin fixation. The above considerations have been added to the manuscript introduction and discussion (lines 51-54 and lines 396-423). Please also see reply 13 for more comments on RNAseq using FFPE tissue.

- 2) that a transformation of methylation data into RNA space provides additional value (which is not shown).

Reply 10: we believe that MethCORR provides several advantages over raw DNA methylation data as:

- 1) For the last decade, numerous studies correlating RNA profiling to cellular phenotypes have generated a wealth of knowledge on the relationship between cellular gene expression and cellular (cancer) biology. This has led to the establishment of many databases and tools, which uses transcriptional profiles to characterize and derive biological understanding from clinical samples. Such tools include subtype classifiers e.g. CMS and CRIS classification, Gene Set Enrichment Analysis (GSEA), prognostic signatures etc. Similar tools are not readily available for raw DNA methylation data, which makes it very attractive to many researchers to convert raw DNA methylation data into a gene-centered format that is compatible with the popular tools and databases. In the present manuscript, we illustrate the power of this gene-centric strategy. Using DNA methylation-derived MCS and iRNA scores for each gene we perform: (1) subtype discovery and classification of samples using the CMS and CRIS classifiers that require RNA expression data as input (Figure 1+2), (2) molecular characterization using GSEA based gene-set analysis using numerous transcriptionally-defined gene sets (Figure 2), (3) biological interpretation of transcriptionally defined gene sets including prognostic RNA expression signatures (Figures 3-4). In addition to presenting the above results, we have highlighted these considerations in the introduction and discussion in the revised manuscript (line 66-68 and line 330-331).
- 2) We also want to highlight that a major strength of our MethCORR approach is that it allows multiple layers of information to be derived from a 450K/EPIC profiling experiment, namely iRNA expression and MCS estimates, DNA methylation estimates, copy number estimates (calculated from the beadchip intensity raw data) and a MethCORR map. This can improve the cost-effectiveness of molecular characterization, as illustrated for CRC in this study. E.g. in Figure 2a (right panel) and supplementary

figure 2f we combine gene expression information (subtypes, ESTIMATE Stroma and Immune score), copy number information (CIN Score) and methylation information (DNA methylation score), which are all derived from the same illumine beadchip analysis. This makes MethCORR attractive for analysis of archival FFPE material e.g. during biological characterization, biomarker discovery and validation. We have now highlighted this added value obtained by performing MethCORR on 450K/EPIC methylation data in the revised manuscript (line 66-68 and line 332-336), which extends beyond transforming DNA methylation data into RNA space.

Additionally, while this reviewer had not previously known of this strategy to infer RNA levels from methylation data, it is not novel. There are several publications doing it.

Reply 11: we fully agree that the concept of trying to infer RNA expression from DNA methylation is not novel, nor was it the primary aim of this manuscript. In fact, imputed RNA scores (iRNAs) are solely presented in Figure 1 and not used for the subsequent analysis in Figure 2-5. Our approach is fundamentally different from earlier studies as it provides a MCS-based framework for uniform characterization and prognostication of CRC FFPE and fresh-frozen samples (in addition to impute RNA expression). In addition, we use expression-correlated CpG sites to associate gene expression with particular cell types (Figure 3 and 4). This is used to interpret published, prognostic RNA signatures and derive simpler, DNA methylation-based PCR biomarker assays that perform well in archival FFPE tissue samples (figure 5). This is discussed in reply 27 and we have cited other studies that impute RNA expression or identify expression correlated CpG sites in the introduction and discussion (lines 71-73, 344-346 and 353-357).

Finally, several technical concerns with the actual modeling and validation procedures themselves were identified, and these are sufficiently large to make downstream interpretation of biological findings difficult.

Reply 12: The majority of the comments by reviewer 2 addresses the imputation of RNA scores (iRNA scores) from DNA methylation via MCSs. We have addressed these concerns (see points A-L below and our replies 15-25) and are very grateful to the reviewer, as this has improved our modelling of RNA expression from MCSs (Figure 1). We would like to highlight that it is the MCSs themselves, and not the modelled iRNAs, that were used for the downstream analysis in Figure 2-5. Still, in the revised manuscript, we have recalculated all results in figure 2-5 after exclusion of 2722 poorly modelled genes. This did not affect the already presented results and conclusions from our downstream analysis of CRC subtypes and prognosis (Figures 2-5).

1. FFPE RNA-Seq

The authors suggest that FFPE material is not amenable to high-quality RNA, and that this precludes robust profiling (lines 58-64). At this point, it is increasingly difficult to support this claim, as there are literally hundreds of papers reporting robust RNA-Seq on FFPE material. And of course there are standard kits that claim to support such analyses: www.illumina.com/science/education/ffpe-sample-analysis/low-quality-ffpe-RNA-seq.html. As quick examples, consider PMIDs: 29862382, 25495041, 31058252. It simply isn't possible to support the idea that methylation profiling of FFPE material is significantly superior to RNA profiling of it. Both are possible, but both can fail in surprising ways, some of which appear to be a function of the specific fixation procedure and reagents, others of which are correlated to age, and still others of which are unknown. The authors use this as the motivation for the study, and this comprises a major weakness.

In some sense, they are tackling a problem which is not current, and for which there is little evidence that methylation data will be universally superior to RNA data.

Reply 13:

We agree with the reviewer that several studies have reported that it is possible to perform biologically meaningful analyses on FFPE-samples. Concurrently, we have also moderated our revised manuscript to better reflect this view and cited publications performing comparative analysis of RNAseq in FF and FFPE tissue (e.g. lines 50-54 and lines , L 396-412). Still, we very respectfully disagree with the reviewer that the problem of poor quality of RNA sequencing in archival FFPE tissue is not a current problem for the following reasons:

1) Reviewer 2 is correct that several companies/products claim to be able to produce RNA-seq. data from FFPE tissue, and that research groups have described generation of FFPE RNA-seq data. This documents a great interest among researchers for using the rich source of FFPE archival samples to perform translational studies. This is also a key motivation for our study. However, simply getting gene expression values does not mean that sequencing results are robust/accurate enough to support integrative analysis of FFPE and FF samples as shown here. E.g. in the publication PMID 31058252, cited by reviewer 2, thirty-eight FFPE samples were accessed by RNA-seq, yet no matched fresh-frozen samples were included for comparison. Therefore, the fidelity of the FFPE RNA-seq vs. fresh-frozen RNA-seq is not possible to evaluate, which is true for many similar studies.

2) Some studies have reported good correlation between RNA sequencing performed in paired FF and FFPE samples, including ourselves (PMID: 24878701). Despite this seemingly good performance, the integration of RNA-seq results from fresh-frozen and FFPE tissues is difficult, because when compared in principal component analyses (PCA) the fresh-frozen and FFPE samples cluster distinct and according to preservation type. We have shown this in Supplementary Figure 1q for TCGA COREAD samples and this observation is also confirmed by us previously (PMID: 24878701) as well as the article (PMID 29862382, cited by reviewer 2). As discussed in reply 9, a dramatic shift in RNA profiles upon formalin storage is reported in PMID:31061401, which may explain the above clustering results. In contrast, samples primarily clustered according to phenotype (i.e. subtype, patient) rather than preservation type when PCA analysis was performed using MCSs (Supplementary Figure 1q and 2e).

3) Many reports of successful RNA sequencing in FFPE tissue, including the two examples provided by reviewer 2 (PMID: 29862382, 254950410), is performed in samples that have been subjected to optimal fixation protocols or have been stored for only short time. E.g. samples used in PMID: 29862382 exhibited optimal fixation conditions, short storage time (8-264 days), and much higher RNA quality (average DV200 values >0.79) than seen for typical archival FFPE samples. The authors acknowledge that “Limitations to our study include small sample size., optimally short time to fixation of tissues and possibly as a result a modest degree of degradation of FFPE samples” (DV200 ranges from 65% to 85%). Similarly, the two FFPE samples analyzed in PMID 254950410, cited by reviewer 2, were only stored for “*about two years*”.

4) A very recent and larger study evaluates FFPE RNA-seq. in older FFPE tissue samples more reminiscent of a “real life situation” and report that they exhibit much lower RNA quality (DV200 could not be measured in 14 of 67 samples, whereas the average DV200 for the remaining samples were 0.17). The authors conclude that merely “60% of samples yielded data that enabled gene expression quantification” and further analysis of seven pairs of biological replicate samples revealed an intra-tumor expression correlation efficiency of “0.7-0.8” for four pairs and <0.22 for the remaining three pairs. Albeit, the authors are still encouraged by these results, they conclude that “There is no denying that there are technical and quality limitations for FFPE RNA-seq data”. Similarly, in PMID: 28122052 only 3 of 11 samples yielded informative RNA sequencing results. We believe that it is highly unlikely that the RNA expression profiles in these two latter studies would support robust subtype discovery and characterization as we demonstrate for FFPE tissue samples and MCSs in this manuscript.

5) In our matched RNA-seq and DNA methylation derived iRNA comparisons of nine COREAD samples, we observed that FFPE RNA sequencing produced less concordant expression profiles than iRNA when compared

to RNA-sequencing from matched Fresh-frozen tissue (Figure 1g). Furthermore, we found that samples with the lowest correlation between RNA expression and iRNA scores had lower quality scores (RIN scores) than samples with high correlations (whereas HM-450K data quality did not differ; new Supplementary Figure 1o). This suggests that RNA-seq expression profiles are influenced by RNA degradation even in fresh-frozen samples, which is in line with previous observations including PMIDs: 24885439, 28446590. Much of the observed variation is likely rooted in variation in the quality of the extracted RNA, however, variation in RNA quality are both difficult to access and control for in FFPE RNA studies (PMID: 28122052).

6) The FFPE RNAseq publications in the CRC field all describe small to medium sized proof of principle studies. There are no publications of large clinically relevant cohorts. Given the easy access to nearly unlimited numbers of attractive, archival samples, then such publications would have existed if FFPE RNA-seq was robust and reliable.

As described in reply 9, we are encouraged that the MethCORR strategy allows robust genome-wide characterization of “real-life” samples using a well-established DNA platform and does not seem to require highly specialized protocols for sample collection, quality assessment and profiling. In this regard, the human methylation beadchip technology is well-proven, robust in FFPE samples, and popular with many public datasets available (e.g. is employed by the TCGA project).

2. TWAS

The core of the method is learning a methylation-to-RNA model for each gene. This is directly analogous to a TWAS study, and it is surprising that the authors do not draw this comparison, nor outline the similarities and difference between the two.

Reply 14: In line with the main aim of the manuscript, we have focused on CRC biology and prognostication more than comparison to genotype-based methods for RNA imputation, but now having become aware of TWAS we have gladly acknowledged this analogy early in the discussion (lines 336-341).

3. Model validation

As was the case in many of the early TWAS studies, model validation here is insufficient. Considering the recent reviews and papers in that field would help significantly with the validation here.

3. Model validation

As was the case in many of the early TWAS studies, model validation here is insufficient. Considering the recent reviews and papers in that field would help significantly with the validation here.

Reply 15: We acknowledge the concerns of reviewer 2 and have, accordingly, expanded and improved our modelling of inferred RNA (iRNA) scores from the MCSs for each gene in the revised manuscript. The many improvements have resulted in significant revision of Figure 1, Supplementary figure 1 and of the manuscript text as summarized below:

- 1) We now build and select regression models in discovery set 1+2 samples using 10x10 fold cross-validation. As described below, this analysis has been performed independently in three different datasets
- 2) We now include polynomial models (second to fourth order), in addition to the simple linear models used in the original submission.
- 3) We now select the best performing model for each gene (i.e. simple linear or second to fourth order polynomial models) in discovery set 1+2 samples using residuals as selection criteria (i.e. low Root Mean Square Error (RMSE) values). Modelling/performance metrics (R^2 , RMSE and mean absolute error (MAE) for all models in the discovery set 1+2 are also now systematically reported for each gene in the supplementary tables.
- 4) The performance of selected models for each gene is now evaluated/validated in the independent “validation set 3” (Figure 1a), which was neither used for identification of expression-correlated CpGs nor during building of regression models for each gene. Modelling/performance metrics (R^2 , RMSE and MAE) for all models in the validation set 3 are now also systematically reported for each gene in the supplementary tables.
- 5) We now introduced a stricter criteria for accepting models and only gene models with inter-sample $R^2 > 0.16$ in both set 1+2 and set 3 is accepted as MethCORR genes. Compared to the original submission, implementation of the criteria caused 2722 genes to lose their MethCORR gene status for the UCSC XENA TCGA COREAD cohort, which leaves 11,222 MethCORR genes for downstream analysis in figure 2-5.

Furthermore, to evaluate and validate the MethCORR principle in different CRC datasets we performed the above MethCORR analysis (i.e. genome-wide identification of expression-associated CpGs, calculation of MCSs and regression modelling) three times using three different datasets with available matched RNAseq and DNA methylation data as follows:

1. Primarily, we used the TCGA COREAD DNA methylation and RNAseq datasets acquired from the UCSC XENA database to generate the COREAD MethCORR matrix (Supplementary table 3) and the set of regression models used throughout the manuscripts, unless otherwise indicated (e.g. Figure 1a,1d-f and Supplementary figure 1; regression model performance metrics are given in Supplementary Table 2 and 4; sample IDs in Supplementary Table 1). Hence, this MethCORR analysis constitutes the backbone of the manuscript and is used for the subtype discovery, characterization, MethCORR map, and biomarker analysis in Figures 2-5.

2. We validated that the MethCORR principle is applicable and robust in an independent CRC cohort by repeating all initial steps of MethCORR analysis (i.e. genome-wide correlation, MCS calculation and regression modelling) using matched RNAseq and DNA methylation data from the Danish SYSCOL cohort. The analysis showed that the MethCORR approach generates models with similarly high intra- and inter-sample modelling performance as when originally performing the analysis using the UCSC XENA datasets. The application of the MethCORR approach is described in line 131-138, Supplementary table 5-7; Figure 1e). This validated the generality of our MethCORR approach. However, our analysis also showed a moderate drop in inter-samples R^2 when comparing inferred RNA expression values in SYSCOL samples, using models established in COREAD, to actual RNA sequencing results in SYSCOL samples. Similarly, we found a moderate drop in inter-samples R^2 when comparing inferred RNA expression values in COREAD samples, using the models established in SYSCOL, to actual RNA sequencing results in COREAD samples (Figure 1e and supplementary tables 5 and 7). We speculated that differences in RNA quantification methods between datasets might be the underlying cause of this, which we addressed in the MethCORR analysis described below.

3) Lastly, to evaluate the impact of data normalization methods on regression modelling and to allow analysis of the nine matched Fresh-frozen - FFPE samples in Figure 1g, we repeated all initial steps of the MethCORR analysis (i.e. genome-wide correlation, MCS calculation and regression modelling) using data from the TCGA COREAD samples acquired via NCI genomic data commons (GDC) database (Supplementary table 8-11). Similarly to the UCSC XENA dataset above, this GDC dataset also encompass RNA expression and DNA methylation data from the TCGA COREAD samples, however, the original data is processed/normalized differently in the two datasets. This allows us to evaluate the impact of data processed/normalization on the

performance of MethCORR models. Our analysis showed that MethCORR iRNA-RNA intra-sample R^2 between the two datasets were not lower than what was attributable to applying two different normalization strategies to the same samples. This is now shown in Supplementary Fig. 1m and introduced in lines 138-143.

Due to these many changes and improvements, we have expanded Figure 1A, updated Figure 1, significantly revised Supplementary Figure 1, revised the manuscript text in the result section (lines 90-143), and expanded the materials and methods section (lines 489-545). Also, we now present modelling metrics, RMSE, R^2 and MAE, in the supplementary tables for all inter-sample (gene-specific) and intra-sample (per-sample) regression analysis: Supplementary table 2 and 4: inter-sample and intra-sample modeling metrics in the UCSC XENA TCGA COREAD cohort, respectively. Supplementary table 6 and 7: inter-sample and intra-sample modeling metrics in the SYSCOL cohorts COREAD FF1, respectively. Supplementary table 9 and 10: inter-sample and intra-sample modeling metrics in the GDC TCGA COREAD cohort, respectively.

Key problems with the modeling and validation are:

- a) The models fail to adjust for critical variables like sex, ethnicity and age which are known to be methylation-associated.

Reply 16: Originally, we did not include age, ethnicity and sex as independent variables in our modelling because the effect of cancer on the genome-wide DNA methylation levels is known to be much more pronounced than these factors. However, to address the concerns raised by reviewer 2, we have performed regression modelling of iRNA scores (from the MCSs as before) and included gender and age as independent variables. The addition did not lead to any significant improvements in model performance, as evaluated by R^2 and RMSE, probably because the effects of gender and age are negligible compared to the large effect of the cancer (Supplementary Figure 1g and lines 114-116). We expect the same result for ethnicity, however, ethnicity status is not available for the FF1 cohort in the phenotype data downloaded from the XENA database and as all patients in the FF2 (SYSCOL) cohort are Caucasians (no diversity in ethnicity exist), ethnicity cannot be meaningfully tested in our study.

While adding gender to the regression models, overall did not improve the derived iRNA scores, gender obviously dictates expression of a range of genes. This is well captured by our MethCORR approach during CpG site selection for each RNA (the initial correlation of each of the 400.000 CpG site to each RNA used to

subsequently calculate MCSs). Here CpG sites with gender-specific differences in methylation will be linked with RNAs that also exhibit clear gender-specific expression. To illustrate this, we now show that iRNA scores from the archetypical female-specific XIST RNA and chromosome Y-bound transcript ZFY both clearly separate male from female patients, as would be expected, in Supplementary Figure 1h and line 116-119)

b) The iRNA scores shown in Figure 1d do not appear to show uniform error, but instead to be associated with abundance level

Reply 17: We fully agree that the error between iRNA and RNA is associated with abundance level. However, this is completely expected in RNA profiling studies where it is a typical pattern that detection fidelity is poorer for lower expressed genes than higher expressed genes. The pattern is also seen in similar plots comparing fresh-frozen RNA-seq and FFPE RNA-seq from matched tumors (e.g. in Figure 5 in the manuscript by Hedegaard *et al.*; PMID: 24878701). Hence, MethCORR cannot model RNA expression with greater accuracy than provided by the fidelity of the underlying RNA-seq analysis. I.e. if the underlying RNAseq does not exhibit uniform error then iRNA scores will likely also not. To illustrate this impact of RNA quantification on iRNA scores, we have introduced a comparison of iRNA scores for COREAD samples (set 3) where the underlying RNA-sequencing data was normalized using two different approaches before MethCORR analysis (data acquired for the same TCGA samples via the UCSC XENA database or via the NCI GDC database; The two databases are using different normalization approaches; see reply 15). In Supplementary Figure 1m, we show that iRNA scores achieve the same correlation R^2 -values in iRNA between datasets as the R^2 -values between RNA sequencing data between the two datasets.

c) There is no gene-specific assesment of model fit

Reply 18: As described in reply 15 we have dramatically revised our modeling of RNA from MCSs and now provide inter-samples (gene-specific) assessment of model fit (R^2 , RMSE, MAE) for the MethCORR analysis of the three independent XENA UCSC TCGA COREAD cohort dataset (supplementary Table 2), SYSCOL cohort dataset (supplementary Table 6) and GDC TCGA COREAD cohort dataset (supplementary Table 9). We also provide intra-sample (sample-specific) assessment of MethCORR model fit (R^2 , RMSE, MAE) for the three

cohorts (XENA COREAD in supplementary Table 4; SYSCOL in supplementary Table 7; GDC COREAD in supplementary Table 10).

d) Model that fit poorly appear to be included in downstream analyses

Reply 19: As described in reply 15 we have dramatically revised our modeling of RNA from MCSs and have now introduced stricter criteria for accepting models: only genes with models with intra-sample $R^2 > 0.16$ in both discovery set 1+2 and in the independent set 3 are accepted as MethCORR genes. Compared to the original submission, these changes caused 2722 genes to lose their MethCORR gene status, and leaves 11,222 MethCORR genes for downstream analysis. The exclusion of poorly modelled genes did not affect the already presented major results and conclusions from our downstream analysis of CRC subtypes and prognosis (Figure 2-5).

e) Per-sample correlations are much less important than per-gene correlations across a cohort

Reply 20: We agree. However, for the iRNA-RNA correlations we wanted to make our results directly comparable to results from previous publications comparing fresh-frozen and FFPE RNA-seq. In these publications, the use of per sample correlation were prevalent (e.g. in PMID: 24878701, 29862382, 25495041). In the revised manuscript, we now provide both per-gene and per-sample correlations for all analysis in the Supplementary Tables (see reply 18).

f) We are not shown the distributions of correlations across genes, making outlier analysis impossible

Reply 21: All inter-sample (gene-specific) correlations are now available in the Supplementary Tables for identification and analysis of outliers (please see reply 18).

g) All values are reported as Pearson's correlations, when there is no evidence to believe these relationships are linear (and indeed, they clearly are not in some cases, e.g. Figure 1e shows definite non-linearity)

Reply 22: To address the situation where the intra-sample relationships are not linear we now report Spearman's rho (rank correlation) for all intra-sample correlations, in addition to Pearson's R^2 in the supplementary tables 4, 7, 10 and 12. As part of the revision, we removed Figure 1e to save space. Figure 1e was just an example from a single sample of the 78 samples that are now included in the new Figure 1e. All model performance metrics, including those from the sample from the old Figure 1e, are now included in the new supplementary table 4. Also, the modelling of iRNA scores from MCS now include polynomial models in addition to simple linear models (please see reply 15).

h) Throughout Pearson's R is reported, but the meaningful metric here is R² (or some other metric of percent variance explained). The correlations of 0.8-0.9 are much less impressive in that space, with values 0.65-0.81, which means a third to a fifth of variance, is not explained

Reply 23: We now report R² values for all inter-sample (gene-specific) and intra-sample (sample-specific) correlation analysis (see reply 18). In our view, the observed intra-sample correlations observed are quite encouraging. It would have been highly surprising if DNA methylation would be able to predict RNA expression with much greater accuracy. It is important to remember that RNA transcript levels are affected by many other things than DNA methylation. There are also technical explanations, including: 1) low fidelity of the RNA expression quantifications by fresh-frozen RNA-seq, particularly for lowly expressed genes (see reply 17). 2) differences in pre-analytical sample handling. Remember that different RNA preparations are used for fresh-frozen and FFPE tissue. 3) That the used methylation analysis was not exhaustive, it "only" addressed 450K positions in the genome. We are satisfied that the correlations between iRNA in FFPE tissues and RNA-seq in matched FF samples have the same or higher R² than was obtained by RNA-seq on matched FF and FFPE samples (Figure 1g). The obtained iRNA-RNA R² values, passes the threshold previously reported for calling an FFPE RNAseq experiment successful (e.g. PMID: 24278466).

i) Beyond the choice of Pearson for correlations, and R vs. R² as a metric, a bigger problem is that correlations aren't the meaningful metric at all. Correlations are giving rank-order, but most RNA abundance analyses (indeed including the many done downstream of this work here) use continuous values not rank-ordered ones. As a result, residuals and similar metrics of deviation from truth are the key values. Those are not reported systematically, and we do not see their distributions

Reply 24: We now systematically report information of residuals, in the form of RMSE and MAE, for all models in the supplementary tables, which allow evaluation of their distributions (please see reply 15).

In agreement with the higher intra-sample correlations (Pearson R² and Spearman's rho) between iRNA scores (FFPE) vs. RNAseq (FF) than for RNAseq (FFPE) vs. RNAseq (FF) in the nine TCGA patients with matched data from FF and FFPE tissue (Figure 1g and supplementary table 12), we now also observed that RMSE values are indeed correspondingly lower for comparisons between iRNA (FFPE) vs. RNA (FF) than for RNA (FFPE) vs. RNA (FF; now included in the revised Figure 1g).

j) There is little to no analysis of the features of consistently well- and consistently poorly-modeled genes

Reply 25: In the revised manuscript, we have now included several analysis of poorly- and well-modelled genes in Supplementary Figure 1b and i-l and introduced corresponding text in the results section (lines 120-126).

k) Line 128, the authors attribute poor sample-wise correlation to low-quality RNA, but do not demonstrate whether these samples have equivalently low-quality methylation data.

Reply 26: We have included an analysis that show no significant difference in the number of failed CpG probes (detection P values >0.05) between top 50 samples with highest and lowest squared correlation, R^2 (Supplementary Figure 1o, lines 150-153). Furthermore, we have now also removed CpG sites from the MethCORR Matrix with high detection p values in FFPE samples (cohorts FFPE1 and FFPE2) to minimize the impact of low-quality methylation data on the MCSs (lines 521-523).

l) The core idea here is not novel, for example PMID:27897002, 29581450 did a simpler version of the modeling, while PMID:31106051 did a much more sophisticated version. Thus the core novelty of the methodologic approach is unclear.

Reply 27: As stated in reply 11, we fully agree that the concept of trying to infer RNA expression from DNA methylation is not novel and we have cited several previous publications in the introduction and discussion, including the papers mentioned by reviewer 2 (e.g. in lines 343-357). Importantly, inferring of RNA expression was not the primary aim of this manuscript: Imputed RNA scores (iRNAs) are solely presented in Figure 1, whereas it is rather the novel MethCORR methylation scores (MCSs) and MethCORR map, which were used for subtyping, characterization and establishment of DNA-based biomarkers by interpretation of expression-correlated CpGs (Figures 2-5). Hence, our approach is fundamentally different from the studies mentioned by reviewer 2, as it provide a novel MCS/MethCORR map-based framework and demonstrate that it allows uniform characterization and prognostication of CRC FFPE and fresh-frozen samples (in addition to impute RNA expression), as indicated by the manuscript title.

We have now extensively modified the manuscript (introduction, results and discussion) to better highlight these novelties of our approach:

- 1) We include additional text (lines 171-177 and 363-377) and two Supplementary Figures (2a-b) to illustrate the rationale and motivation for creating the novel expression-correlated MCS for each gene and use this as an alternative to RNA expression profiles during subtype discovery. This data illustrates that the MCS is a

more “cancer-focused” metric than RNA and less influenced by the transcriptional variation found in normal mucosa biopsies, as described below in reply 28.

- 2) We illustrate that the MCSs support robust subtype discovery, classification and characterization in both fresh-frozen and FFPE cohorts using popular bioinformatics tools (Figure 2). Also, we show that MethCORR allows three layers of molecular information to be extracted from a single DNA methylation BeadChip microarray, namely a gene expression profile (MCSs or iRNAs), a methylome profile (450K/EPIC profiles) and a chromosome copy number profile, calculated from the methylation arrays signal intensity (line 332-341).
- 3) We illustrate that our novel MethCORR map can, by integrating RNA expression and DNA methylation data, be used to describe sources of inter-tumor heterogeneity, to interpret RNA expression signatures including prognostic signatures, and to derive DNA-methylation based biomarkers, which can be validated by PCR-based strategies (Figures 3-5).
- 4) We demonstrate that MethCORR offer an alternative, and often superior, route to acquire RNA expression information (i.e. iRNAs) for 11,222 genes when high quality RNA is not available, such as in archival FFPE samples (Figure 1g). Notably, we achieved this by identifying genome-wide correlations between the expression of each RNA and each CpG included in the UCSC XENA TCGA COREAD dataset and used this information to establish a MCS for each gene from which RNA expression can be modelled. This is different to the studies highlighted by reviewer 2 above, which have included only CpG sites near the gene (PMID: 31106051) or only top CpG sites with high variability in DNA methylation (≤ 500 in total; PMID: 27897002, 29581450) for identification of expression-correlated CpG sites. Likely for this reason, we find that gene expression can be modelled for $\sim 10x$ more genes with similar R^2 using our genome-wide strategy than e.g. PMID:31106051. Also, we provide new analysis of well-modelled genes (supplementary figure 1b, i-l) and expression-correlated CpGs (supplementary figure 3a-b).

We hope that the reviewer 2 may now appreciate that both the MethCORR approach and the outcomes we achieve using it (establishment of a cancer-centric MCSs, MethCORR map, CRC subtype discovery, characterization, prognostication and DNA methylation biomarker development) are fundamentally different and more comprehensive than reported in earlier studies of RNA imputation alone.

- M) The actual MCS is something like the average beta-value (scaled to give symmetry). It is unclear why this is reasonable, and why better-correlated probes are not given higher weights

Reply 28: The underlying rationale for the MCS design is that the expression of a particular gene typically will be positively associated with the methylation pattern of some CpGs and negatively associated to others. E.g. promoter-associated CpG sites may often be negatively associated with RNA expression (unmethylated during expression), whereas CpG sites within the gene body will exhibit an opposite pattern (methylated during expression). Analogously, in biopsies with great cellular heterogeneity, like those that are obtained from CRC tumors, the expression of cell type-specific RNAs (such as the T cell-specific CD3E) will be associated with T cell-specific methylation patterns, including both unmethylated and methylated sites (both sites that are local to the gene and distant from the gene). As methylation β -values ranges between 0 (unmethylated) and 1 (fully methylated), the relative content of T-cells in a sample may therefore be determined as: 1) the β -value of sites with T cell-specific methylation or 2) “1 minus the β -value” of sites with T-cell specific “non-methylation”. This mimics the situation with promoter and gene body associated probes. This principle is also used in “Epigenome-Wide Association Studies (EWAS; PMID: 28193155)” to evaluate the proportion that individual cell types constitute within complex samples.

Therefore, as this is a common principle, we chose to calculate the MCS as an average between β values for expression-methylated probes and “1 minus β -values” for expression-“nonmethylated” probes (Figure 1b).

Notably, due to this design, the MCS will exhibit greater variation for genes, whose expression-associated CpGs exhibit the greatest variation in DNA methylation β -values. Hereby, cell types with great abundance and variation in the CRC samples (such as types of epithelial cancer cells) will exhibit greater variation in MCSs between samples than cell types with lower abundance (such as immune, endothelial cells etc.). Therefore, employing MCSs will expectably emphasize cancer cell-related traits during subtype discovery and thereby help address the major concern that RNA-based subtyping has been reported to be severely affected by the stromal content of the tumor biopsy (PMID:27151745). In the revised manuscript, we now include additional text (lines 171-173 and 359-373) and two supplementary figures (2a-b) to illustrate these motivations for creating and using the expression-correlated MCS score.

In regard to weighing of CpG sites, we have now performed modelling of RNA expression directly from the β -value of the CpG sites included in the MethCORR matrix for each gene using multiple linear regression and compared it to our MCS-based models for RNA imputation. Our results show, that the multiple regression models and the MCS-based models have equal performance, which suggests that the CpG weight established during multiple linear regression modelling does not improve our predictions (Supplementary Figure 1f; lines 111-113). We have therefore focused our manuscript on modelling RNA expression using the single MCS

variable as input, as MCS scores also form the basis for the results generated throughout the remaining manuscript (Figures 2-5).

Minor

* the critical MCS value is not defined in methods, but placed in a figure, which is a poor decision (placing it in both is fine)

Reply 29: We have now added the definition of the MCS to the material and methods section.

REVIEWERS' COMMENTS:

Reviewer #2 (Remarks to the Author):

This a revised version of the manuscript "MethCORR: DNA Methylation-based Characterization, Classification and Prognostication of Colorectal Cancer using Archival Formalin-fixed, Paraffin-embedded Tissue" by Bramsen and colleagues.

I remained with the idea that this is a great manuscript.

Authors answered all comments from reviewers.

Reviewer #3 (Remarks to the Author):

The authors have addressed all my technical concerns. I still have major concerns about novelty and utility, but I'm happy to leave those as an editorial decision.

REVIEWERS' COMMENTS:

Reviewer #2 (Remarks to the Author):

This a revised version of the manuscript " MethCORR: DNA Methylation-based Characterization, Classification and Prognostication of Colorectal Cancer using Archival Formalin-fixed, Paraffin-embedded Tissue" by Bramsen and colleagues.

I remained with the idea that this is a great manuscript.

Authors answered all comments from reviewers.

Reviewer #3 (Remarks to the Author):

The authors have addressed all my technical concerns. I still have major concerns about novelty and utility, but I'm happy to leave those as an editorial decision.